# Increased Indian Ocean-North Atlantic Ocean warming chain under greenhouse warming

Young-Min Yang [1,2], Jae-Heung Park [3], Soon-Il An [3,4✉], Sang-Wook Yeh [5], Zhiwei Zhu [1], Fei Liu [6], Juan Li[1], June-Yi Lee [7,8] & Bin Wang [1,9]

Over the past half a century, both the Indian Ocean (IO) and the North Atlantic Ocean (NA) exhibit strong warming trends like a global mean surface temperature (SST). Here, we show that not only simply as a result of increased greenhouse gases, but the IO-NA interaction through atmospheric teleconnection boosts up their warming trends. Climate model simulations demonstrate that the IO warming increases the NA SST by enhancing the longwave radiation through atmospheric teleconnection, subsequently, the warmer NA SST-induced atmospheric teleconnection leads to IO warming by reducing evaporative cooling with weakened surface winds. This two-way interaction (i.e., IO-NA warming chain) acts as positive feedback that reinforces warming over both ocean basins. The Pacific Ocean is partly involved in this warming chain as a modulator in an interdecadal timescale. These results highlight the importance of understanding ocean-basin interactions that may provide a more accurate future projection of warming.

[1] Key Laboratory of Meteorological Disaster, Ministry of Education (KLME)/Joint International Research Laboratory of Climate and Environment Change (ILCEC)/Collaborative Innovation Center on Forecast and Evaluation of Meteorological Disasters (CIC-FEMD), Nanjing University of Information Science and Technology, Nanjing 210044, China. [2] State Key Laboratory of Numerical Modeling for Atmospheric Sciences and Geophysical Fluid Dynamics, Institute of Atmospheric Physics, Chinese Academy of Sciences, Beijing 100029, China. [3] Division of Environmental Science and Engineering, Pohang University of Science and Technology, Pohang 37673, Korea. [4] Department of Atmospheric Sciences and Irreversible Climate Change Research Center, Yonsei University, Seoul 03722, Korea. [5] Department Marine Sciences and Convergent Technology, Hanyang University, ERICA, Ansan, South Korea. [6] School of Atmospheric Sciences Sun Yat-Sen University, Key Laboratory of Tropical Atmosphere-Ocean System Ministry of Education, and Southern Marine Science and Engineering Guangdong Laboratory, Zhuhai 519082, China. [7] Research Center for Climate Sciences, Pusan National University, Busan, South Korea. [8] Center for Climate Physics, Institute for Basic Science, Busan, South Korea. [9] Department of Atmospheric Sciences and International Pacific Research Center, University of Hawaii, Honolulu, HI 96822, USA. ✉email: sian@yonsei.ac.kr

The Indian Ocean (IO) and North Atlantic Ocean (NA) have experienced significant warming trends over the past few decades[1–7]. IO warming affects East Africa, East Asia, El Nino-Southern Oscillation (ENSO), and the North Atlantic climate[8–11]. Furthermore, NA has had significant impacts on South America, the Sahel, Atlantic hurricanes, and the U.S. and European climates[12–15].

Previous studies have discussed how the IO and the NA influence each other[1,7,16–19]. Warming of the IO induces NA warming via atmospheric teleconnection and oceanic processes[1,16]. For instance, IO warming has changed the recent NA climate by influencing tropical rainfall and atmospheric heating[7,17,18] through the inter-oceanic transfer of seawater that increased the sea surface temperature (SST) in the NA[19]. Moreover, warming of the NA induced IO warming either by increasing surface heat fluxes[20–22] or by generating easterly surface wind anomalies[23–26] that may lead to oceanic heat transport into the IO[27–29].

The above-mentioned studies reported the disconnected effects of the IO and the NA on each other and the interactions between IO and NA are not fully understood. Because SSTs of both IO and NA tend to increase and can influence each other, we hypothesized that significant positive feedback in the SST warming might exist between the two regions. In other words, IO warming contributes to NA warming, and the high SST in the NA, in turn, reinforces further IO warming.

Here, we demonstrate the positive feedback between the SSTs of the IO and NA and reveal a physical process for the formation of this IO-NA warming chain. We conducted a historical experiment using an Earth system model following the Coupled Model Intercomparison Project Phase 6 (CMIP6) protocols while achieving enforced control of SST anomalies over the IO (or the NA) through observations. We also performed an idealized simulation with uniform surface warming or cooling (from $-3\,°C$ to $3\,°C$ at intervals of $1\,°C$) over the IO (or the NA) at a fixed $CO_2$ concentration and found that the NA SST warms up linearly with the given IO SST anomalies; additionally, the IO SST influenced by NA warming also increased monotonically. We elaborate on the mechanisms governing these links that are based on atmospheric teleconnection. Furthermore, we show that under strong anthropogenic forcing, the IO-NA warming chain is strengthened by a warmer mean SST in the Pacific, which may affect the local climate in the midlatitudes.

## Results

**Evidence for the existence of Indian – North Atlantic Ocean relationship.** To quantify the strength of the proposed IO-NA warming chain, we defined the IO SST index (IO index) and NA Ocean SST index (NA index) by identifying the SST averaged over a selected region of interest marked by a square box in Fig. 1a. For both SST indices, the long-term trends were removed and an 11-yr running mean was applied before further analysis was conducted. The IO index showed a significant warming trend ($0.06\,°C$ per decade) from 1950 to 2020 and a maximum interdecadal fluctuation around 1940. From 1970 to 2020, the NA index showed a warming trend ($0.12\,°C$ per decade), with strong multi-decadal fluctuations (Fig. 1d). Local maximum and minimum NA indices were observed around 1940 and 1970, respectively. Interestingly, the warming trends and interdecadal-to-multidecadal fluctuations in the IO and NA indices tend to overlap. We examined the relationship between IO and NA indices using the observed data (1950–2020). The peak of lead-lag correlation between the IO and NA SST was found to occur at 0-lag with a coefficient of 0.8–0.9 (Fig. 1e, red line), indicating that concurrent NA and IO warming may be significant. The correlation coefficient decreased with an increased lag (lead) years

and showed a correlation coefficient of 0.7 at $-8$ year lag and 0.5 at $+10$-year lag. Global warming due to anthropogenic forcing may reflect warming trends in the two indices (Supplementary Fig. 1c); however, global warming cannot fully explain the phase matching in interdecadal-to-multidecadal variations. The patterns in the warming trends of both these regions point to the plausible strong interactions between the two basins. To test our conjecture that IO and NA might serve as mutual pacemakers, we began by studying the lead-lagged correlation using a longer period. It did not change much when we used the 1900–2020 SST data (Supplementary Fig. 2a). We realized that the relative IO warming (RIO; the ratio of IO SST to tropical mean SST) was probably more relevant than the absolute IO warming in terms of its teleconnection impacts[2,17]. We additionally examined the lead-lagged correlation between the RIO and North Atlantic (NA) SST. The peak correlation occurred during 0–2 years and the second peaks were found at $-8$ years and $+10$ years (Supplementary Fig. 2b). The correlation patterns between RIO and NA were observed to be similar to those of IO and NA, but their magnitudes weakened. The temporal evolution of the RIO was also similar to that of the IO (Supplementary Fig. 2c), suggesting that the IO could be an appropriate index to represent its interaction with NA warming.

To further analyze the IO-NA relationship, we examined global SST anomalies and surface winds regressed onto the NA index using observational data from 1950 to 2020 (Fig. 1b). The NA warming pattern resembled a developing positive phase of the Atlantic Multidecadal Oscillation (AMO), and subsequent warming was observed in the tropical IO. In the observation, the warm SST anomalies over the NA generated ascending motions and strong upper-level divergent flows, which are connected to upper-level convergences in the equatorial central and eastern Pacific regions (Supplementary Fig. 3a). The descending motions connected to the upper-level convergence induced easterly anomalies over the central to western Pacific regions (Fig. 1b). Further, accumulated warm SST in the western Pacific likely strengthened the Walker circulation by enhancing the ascending motion over the equatorial western Pacific. The model reproduced observed patterns reasonably well but the divergence in the western Indian ocean is weaker than the observations (Supplementary Fig. 3c).

Similarly, SST and surface wind anomalies associated with IO warming were observed to be linked to strong NA warming (Fig. 1c). The Pacific SST patterns resembled local SST trends. Thus, the warming trend in the Pacific Ocean may be related to both global and IO warming. The ascending motion in the IO generated a descending motion over Africa and propagate to the tropical Atlantic Ocean in the upper level and then induce ascending motion there with the NA warming (Supplementary Fig. 3b). On the other hand, the rising motion over the IO generates a sinking motion over the Pacific region. Because the simultaneous regression did not explain causality, we conducted numerical experiments to illuminate the causes and effects of the IO and NA warmings.

**Indian – North Atlantic Ocean warming chain simulated without anthropogenic effects.** We hypothesized that the IO-NA warming chain could be generated without the intervention of any anthropogenic forcing. To verify this hypothesis, we conducted multiple idealized numerical simulations with fixed CMIP6 preindustrial (PI) forcings to remove the warming effects caused by increased $CO_2$. Under fixed PI external forcings, we conducted a suite of 100-year simulations by using observed IO SST anomalies averaged from 1970 to 2020 (PI_IO + 0 C) and progressively increased SST anomalies by $1\,°C$ (PI_IO + 1 C),

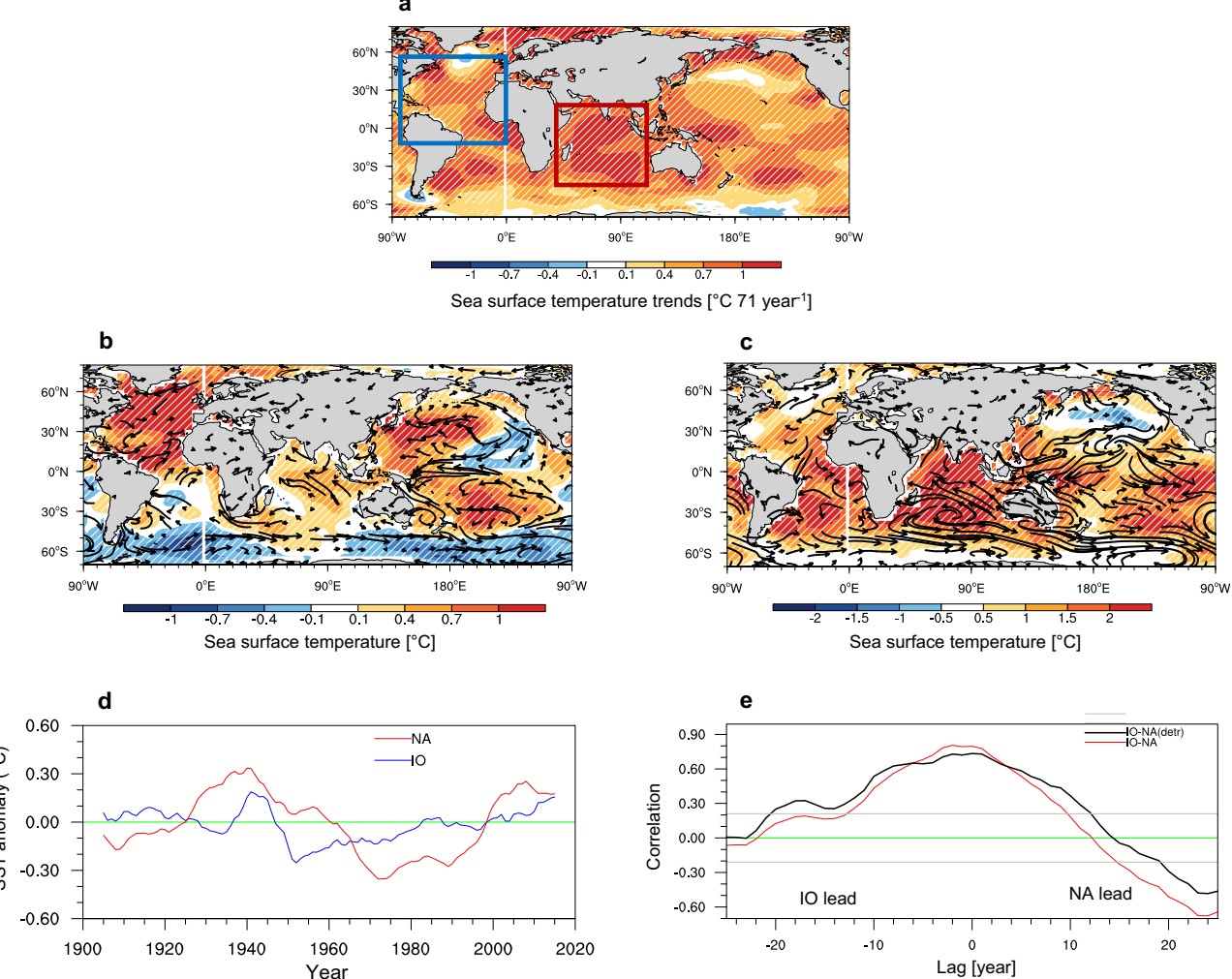

**Fig. 1 Historical warming trends. a** Observed trends (1950–2020) in annual-mean sea surface temperature (SST; units: °C 50 year⁻¹). The red (blue) box represents the region where the Indian (North Atlantic) Ocean index was defined. Observed SST (K) and surface wind anomalies (arrow, m/s) regressed onto (**b**) the North Atlantic Ocean (NA) index (0°–70°N, 80°–0°W) and (**c**) Indian Ocean (IO) index (30°S–30°N, 40°–120°E). The hatched area represents the regressed SST and is significant at the 95% confidence level. **d** Observed time series of the NA (red line) and IO (blue line) indices. **e** Observed lead-lag correlation coefficient between the IO and NA indices. The lag is positive (negative) when the IO leads (lags) and the grey lines represent a 95% significance level. For **b-d**, the 11-year running average data were used for 1950–2020 and long-term linear trends were removed. For **e**, the last 5 years were excluded from the analysis, and the long-term linear trends in the SST data were removed before regression for the only black line.

2 °C (PI_IO + 2 C), and 3 °C (PI_IO + 3 C). We repeated the same experiment with SST anomalies by that reduced by 1 °C (PI_IO-1C), 2 °C (PI_IO-2C), and 3 °C (PI_IO-3C). Additionally, we conducted a similar suite of fixed forcing PI experiments with a nudged SST over the NA region (PI_NA). The model in this study was integrated for 100 years and the 100-year data were utilized for the analysis.

We observed that the NA SST anomalies induced by 1 °C warming over the IO region showed significant NA warming (Fig. 2a). Warming also occurred in the southern subtropical Pacific, equatorial western, and eastern Pacific regions; however, relatively weak warm SSTs prevailed in the northern Pacific. The anticyclonic flows in the south Pacific generated easterly (westerly) anomalies in the western (eastern) Pacific regions, respectively, which may induce a warm SST there.

To understand how IO warming affected the NA SST by atmospheric teleconnection, we examined differences in the temporal evolution of surface heat fluxes between PI_IO + 0 C and PI_IO + 1 C. Note that surface heat fluxes correspond to the impact of IO on NA through atmospheric teleconnection during

most simulation periods. Figure 2b depicts that NA SST increased rapidly, upon IO warming, for approximately the initial 20 years; a constant increase in the NA SST was observed for 50 years. The surface net heat flux was positive all the time (Fig. 2e). The net surface flux adjusted very quickly, within a span of a few years, but slowly decreased with time. The longwave radiative over NA, the largest contributor to the surface net heat flux, increased during the initial 20 years and then decreased slowly (Supplementary Fig. 4).

In the model simulation, the changes in atmospheric circulation are similar to the observed pattern. The strong ascending motion by the warmer SST over the Indian ocean moves to the troposphere of the tropical Atlantic ocean, with sinking motion in the lower level and ascending motion in the middle and higher level (Supplementary Fig. 5a). Meanwhile, the increased cloudiness over the western tropical Atlantic by IO warming (Fig. 2c) may contribute to surface warming by enhanced longwave radiation to the ocean (Fig. 2d). These results showed that the response of NA SST by IO warming is dominated by atmospheric teleconnection with enhanced longwave radiation at the surface.

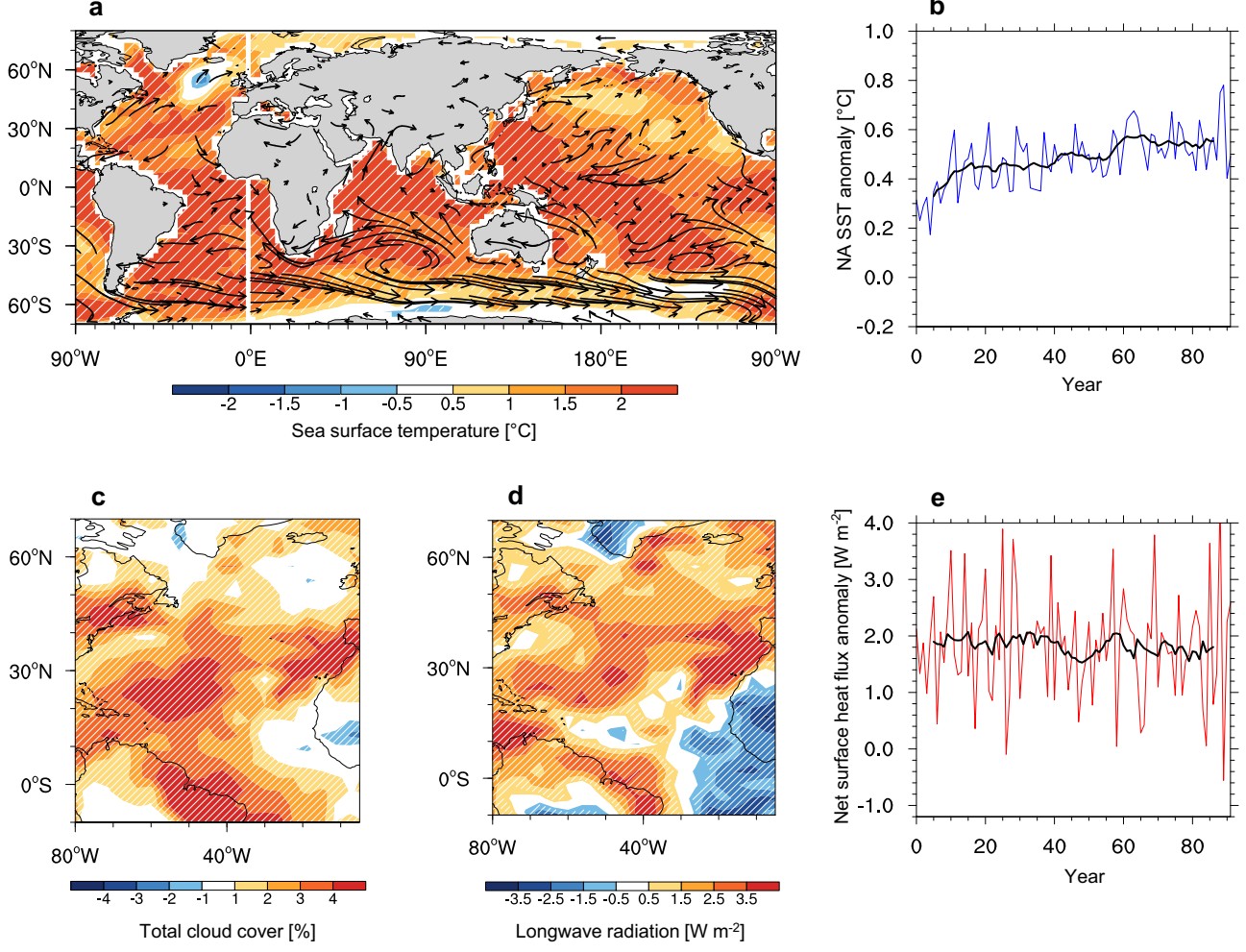

**Fig. 2 Response of North Atlantic Ocean by Indian ocean warming from preindustrial simulations. a** Anomalies in sea surface temperature (SST; unit: °C) and surface wind (vector in units of 0.6 mm per day) from the fixed preindustrial (PI) simulation with observed IO SST anomalies. The anomalies are computed for the 100 years of the PI_IO + 1 C experiment with respect to the pre-industrial simulation (PI_IO + 0, see Methods). The hatched area represents the regressed SSTA and is significant at the 95% confidence, level. **b** Temporal evolution of SST anomalies averaged over the NA region. **c** Anomalies in total cloudiness (%). **d** Anomalies in longwave radiation (W m$^{-2}$). The downward direction is positive (warming). **e** Temporal evolution of net surface fluxes (W m$^{-2}$). The red (or blue) line shows annual mean data and the black line shows an 11- year running mean of the red (or blue) line. 100-year simulated data are used for analysis.

The IO SST and surface wind changes induced by 1 °C NA warming largely resembled those observed in this study and showed significant IO warming (Fig. 3a). Warming also occurred in the northern and southern subtropical Pacific and equatorial western Pacific; however, relatively less SST warming prevailed in the eastern Pacific. The anticyclonic flows in the north and south Pacific is connected to easterly anomalies in the western Pacific and the eastern Indian Ocean (Supplementary Fig. 6a). As changes in the SST and surface winds by NA warming may affect the surface fluxes, we performed a surface heat budget analysis by computing the differences in the four surface heat flux components over the NA region for 1 °C warming (PI_NA + 1 C) and no warming (PI_NA + 0 C). The surface energy changes are mainly controlled by latent heat flux (Fig. 3b), which is about two times greater than all other surface fluxes combined (Fig. 3c). Positive latent heat fluxes were consistent with the IO warming and thus contributed to IO warming. These results suggested that NA warming induces IO SST warming due to reduced wind-driven latent heat flux through modulated Walker circulation in the Indo-Pacific.

We further examined how the NA warming trends were linked to the prescribed IO warming under PI forcing (Fig. 4a, blue line). When the IO warming increased, the NA SST was progressively warmer. The increasing rate of NA SST per 1 °C of IO warming was approximately 0.39 °C. The correlation coefficient between the relative IO warming and the NA response was computed to be 0.98, implying that the response of NA to IO forcings was linear, as per the model simulations. However, the rate of increase of IO SST per 1 °C of NA warming was observed to be 0.33 °C and the correlation between their SST changes was 0.97 (Fig. 4b, blue line). Our results indicate that the IO-NA warming chain can occur through a greenhouse gas forcing-independent mechanism. The historical simulation showed much higher trends of NA warming by IO warming forcing than those observed using the PI simulation. Thus, anthropogenic forcing may intensify the increasing trends in NA warming induced by IO warming.

**Indian – North Atlantic Ocean warming chain evidenced through historical simulations.** To explore whether IO and NA warmings indeed affect each other, we conducted two groups of

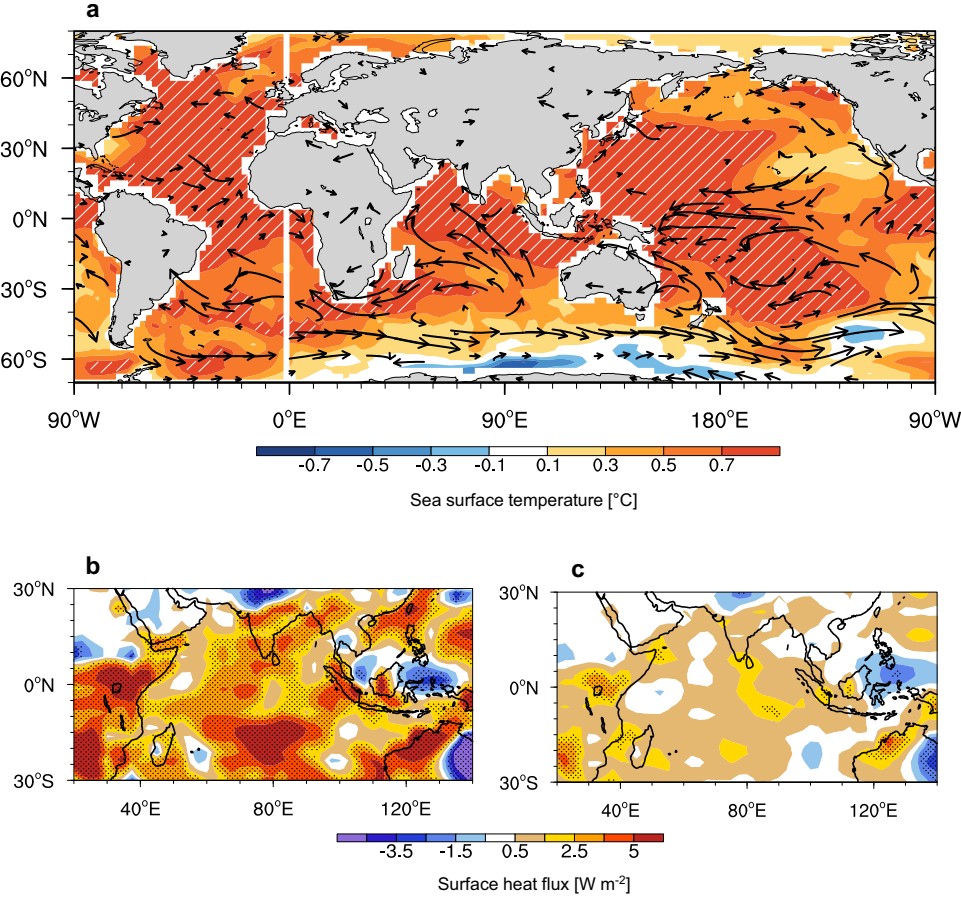

**Fig. 3 Response of Indian ocean to North Atlantic Ocean warming from pre-industrial simulations. a** Anomalies in SST (°C) and surface wind (vector in units of 0.6 mm per day) from the fixed preindustrial (PI) simulation with observed NA SST anomalies. The anomalies are computed for the 100 years of the PI_NA + 1 C experiment with respect to the pre-industrial simulation (PI_NA + 0, see Methods). The hatched area represents the regressed SSTA and is significant at the 95% confidence level. **b** Horizontal pattern of latent heat flux (W m$^{-2}$). **c** combined anomalies of sensible heat flux, solar and longwave radiation (W m$^{-2}$). 100-year simulated data are used for analysis. over the IO region. The downward direction is positive (warming). 100-year simulated data are used for analysis.

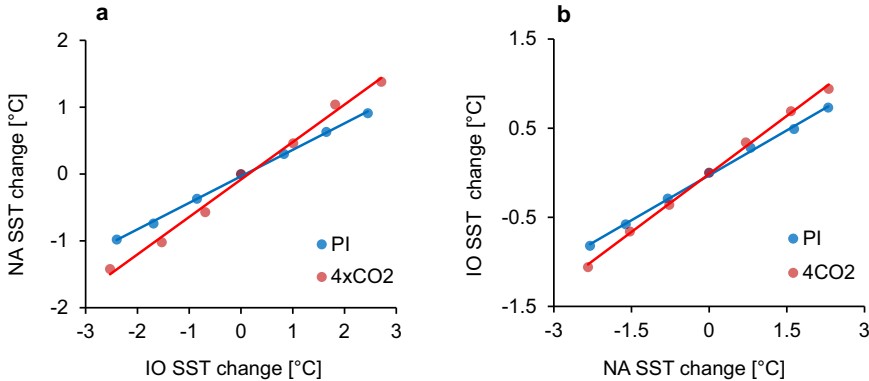

**Fig. 4 Indian Ocean (IO) - North Atlantic Ocean (NA) warming trends derived from preindustrial and greenhouse warming simulations. a** Changes of NA index versus changes of IO index and (**b**) observed NA SST anomalies in different preindustrial (PI) and quadruple CO$_2$ (4CO$_2$) experiments. The NA SST change is defined by the difference of the NA index between the PI simulation with observed SST anomalies and that with relatively 1 °C uniformly warmer SST over the IO. The blue (red) line shows the corresponding linear regression curve. Each circle represents data from one experiment averaged for 100 years.

numerical experiments using an Earth system model (see the Methods section). The first group represented a historical simulation with the prescribed observed IO SST (HIS_OBS_IO). The model was freely coupled in other areas, and external forcings were based on the CMIP6 historical simulation protocol.

Similarly, we conducted model simulations for the other group by nudging the observed SST anomalies over the NA region (HIS_OBS_NA). The model in this study was integrated from 1900 to 2020 with historical external forcing and 71-year data (1950–2020) utilized for the analysis (Supplementary Fig. 6). The

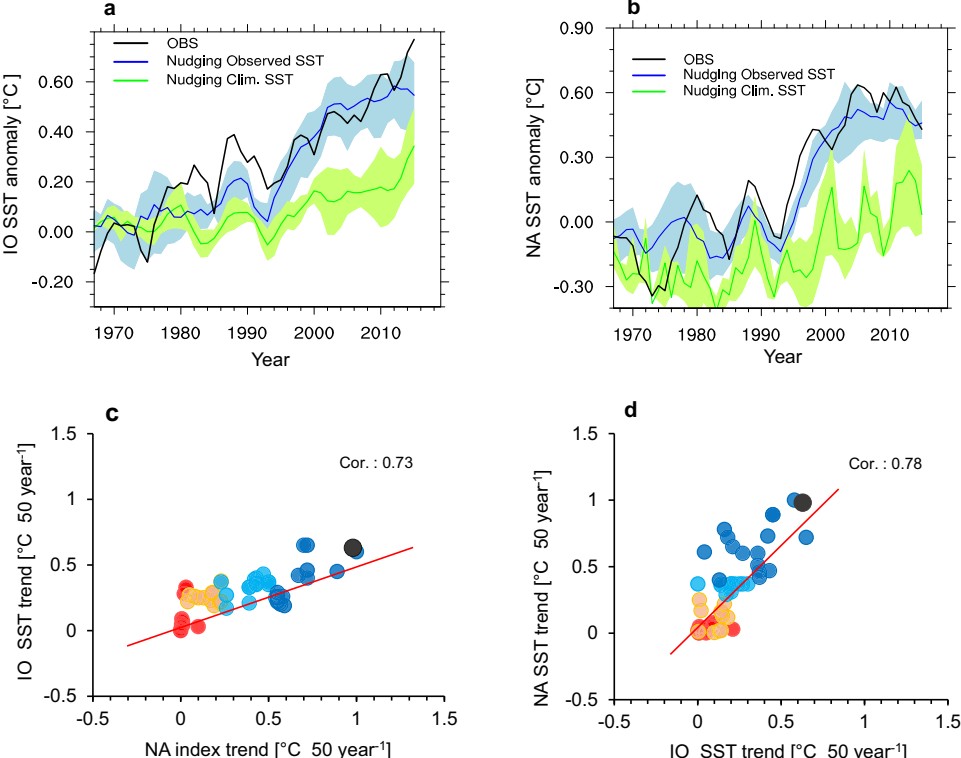

**Fig. 5 Indian Ocean (IO) - North Atlantic Ocean (NA) warming trends simulated via historical experiments. a, b** Historical simulation with observed NA (or IO) SST. Temporal evolution of the IO index derived from historical simulations with the observed SST anomalies (blue) and climatological SST (green) prescribed over (**a**) the NA and (**b**) the IO regions. The black curve denotes the observed IO index. The blue (green) curves represent the model ensemble mean, and the light blue (green) shadings show the ensemble spread. The black lines denote the observations. **c, d** Historical simulation with observed NA (or IO) SST pattern and different trends for 1950–2020 Scatter plots between IO and NA index trends (°C) per 50 years derived from the historical simulations with prescribed (**c**) NA warming trends and (**d**) IO warming trends. Two groups of historical experiments were used for analysis. In the first group, observed NA warming signals were engaged in the model by nudging the observed SST anomalies with various trends for the NA region.

model's capability in the simulation of the observed IO-NA relationship was confirmed by showing the resemblance between the observed regression maps of global SST and surface wind anomalies onto the NA (or IO) for 1950–2020 and those simulated (Supplementary Fig. 6).

The IO Index ensemble simulated by the observed SST anomalies over the NA region (HIS_OBS_NA) captured the observed trends reasonably well for most of the periods (blue line in Fig. 5a). The range of mean for each member from the ensemble was ±0.12 °C (approximate biases of 10–15%), indicating that the impacts of internal variability on the simulated IO SST were moderate; thus, the response of the IO influenced by NA warming was realistic. We re-conducted the same historical simulation, except for the prescribed climatological SST of NA (HIS_CLIM_NA), to eliminate the impact of NA warming on IO warming. The results showed that IO SST warming was reduced by 40–50% (green line in Fig. 5a), indicating that the observed IO warming arose from the remote effects of NA as well as local IO warmings.

For another historical simulation with observed SST forcings over the IO region (blue line in Fig. 5b), the mean SST ensemble over the NA region depicted a realistic temporal evolution, with negative SST anomalies being observed from the 1970s to the 1990s along with rapidly increasing positive SST anomalies after the late 1990s. It was slightly less than that observed, which may be attributed to the systematic mean SST biases of the south of Greenland[30]. The ensemble spread from each member was moderate (±0.26 °C) signifying that all ensemble simulations captured the NA warming trends induced by IO warming

reasonably well. The same historical simulation, but with climatological SST nudged over the IO region, yielded a 50% reduction in NA SST warming (green line in Fig. 5b). This result suggested that approximately half of the NA warming may be attributed to IO warming.

We further examined the sensitivity of the IO (or NA) response to NA (or IO) warming. For this, we conducted two additional groups of model experiments. The first involved historical simulations with observed IO SST patterns and trends (0.01–0.70 °C per 71 years). Different trends were obtained by multiplying different scale factors with the observed values (Supplementary Fig. 7a). The second involved specifying different trends of observed SST anomalies over the NA region (0.01–1.02 °C per 71 years) (Supplementary Fig. 7b). The model was freely coupled in other areas, and external forcings were based on the CMIP6 historical simulation protocol.

Historical runs that nudged various IO (or NA) SST warming (Fig. 5c, d) demonstrated that increasing IO (or NA) SST could lead to a warmer NA (or IO). The IO SST trend forced by the NA SST was smaller than the NA SST trends (by approximately 40–50%). Thus, IO warming may be attributed to a remote NA impact and local surface flux changes. We will study this issue in more depth in our future studies.

**Change in Indian – North Atlantic Ocean warming chain under anthropogenic forcing.** We conducted a suite of 100-year simulations under quadruple $CO_2$ external forcings by using prescribed observed SST anomalies that were averaged from 1980 to 2018 in the IO (4CO2_IO) and then progressively increased

the SST by 1 °C, 2 °C, and 3 °C to determine the effects of continued anthropogenic forcing on the IO-NA warming chain. We repeated the same experiment, but with SST anomalies that decreased uniformly by 1 °C, 2 °C, and 3 °C. Additionally, we repeated 4CO2_IO experiments with observed SST anomalies over the NA region (4CO2_NA).

The NA response to IO warming forcing was linear in the 4CO2_IO experiment. The rate of NA SST increases with 1 °C of IO warming was approximately 0.56 °C, 50% higher than that observed in the PI_IO (red line in Fig. 4a). Under strong anthropogenic forcing, the net surface flux due to IO warming enhanced more than those obtained upon PI simulations. Similarly, the change in IO SST due to NA warming remained linear, with a correlation of $r = 0.98$. The rate of increase in IO SST due to NA (0.39 °C) warming under strong anthropogenic forcing was slightly higher than that in the PI simulations (0.33 °C) (red line in Fig. 4b). Under enhanced greenhouse gas forcing, NA warming generated strong anomalous easterlies in the IO and significantly reduced latent heat fluxes resulting in a warmer SST over the IO regions. These results show that the IO-NA warming chain was intensified by anthropogenic forcing.

**Possible role of Pacific on Indian – North Atlantic Ocean warming chain**. To explore how the Pacific affects IO-NA interactions, we examined the lead-lag relationship between NA (or IO) and the Pacific indices (interdecadal Pacific oscillation, IPO; Pacific Decadal Oscillation, PDO; and decadal ENSO) using the observed data (1950–2020). The IPO is defined as the second EOF (after the global warming mode) of decadal (11 years) low-pass filtered SST[31]. PDO is defined as the leading principal component of North Pacific monthly sea surface temperature variability. The decadal ENSO is defined as the Nino3.4 index of decadal (11 years) low-pass filtered SST. Noted that positive IPO pattern shows El Nino-like SST patterns. Within the decadal time scale (~10 years), the peak of lead-lag correlation between the NA and PDO was found to occur at 3–5 lag years with a coefficient of 0.8–0.9 (Supplementary Fig. 8a, blue line), indicating that NA leads to positive PDO significantly with 3–5 lag years. Similarly, the NA leads the negative IPO phase with 6–10 lag years. The relationship between NA and decadal ENSO is relatively weaker than those of PDO and IPO. The IO leads both PDO and decadal ENSO at 3–5 lag years but there is no significant relation with IPO (Supplementary Fig. 8b). These results showed that both IO and NA warming induces positive PDO with 3–5 lag years, suggesting that the IO-NA warming chain may interact with IO (NA) through PDO.

For the multi-decadal time scale, the positive IPO and decadal El Nino lead to both IO and NA warming at 20 lag years, while IO and NA warming induces negative IPO and decal La Nina phase at 10–20 lag years. This lagged relationship may be attributed to the periodicity of the IPO and decadal ENSO. In the relationship with PDO, the NA leads to negative PDO at 25 lag years but IO has no significant correlation with PDO at a multi-decadal time scale. These results showed that positive IPO may contribute to IO and NA warming with a lag of 20 years and on the contrary, the IO-NA warming could incudes negative IPO with a lag of 10–15 years, suggesting that IPO plays a role in the IO-NA warming chain with multi-decadal time scale. In summary, the Pacific indices could contribute to interdecadal modulation of the IO-NA warming chain, rather than a simultaneous feedback relationship.

## Discussion
In this study, we demonstrated a strong relationship between IO and NA warming (the IO-NA warming chain) and the intensified

IO-NA warming chain due to global warming may affect IO-induced (NA-induced) local climate change in the mid-latitudes (including precipitation, temperature, and fire emission). These may be interesting topics for further studies.

The timescale dependency of high-latitude (50°–70°N) NA SST due to IO warming could be an interesting issue to consider. The horizontal pattern of SST anomalies between PI_IO + 0 C and PI_IO + 1 C experiments show moderate warming over the equatorial and subtropical Atlantic region with relative cooling near 40°–60°N during the initial phase (0–30 years) (Supplementary Fig. 9a). The relatively weak warming in the south of Greenland gradually changed to basin-wide warming during the final phase (70–100 years) (Supplementary Fig. 9b). The enhanced meridional sea surface salinity gradient and AMOC due to IO warming (Supplementary Fig. 10) could increase the northward transport of ocean heat, which subsequently would warm the entire North Atlantic (Supplementary Fig. 9c). However, it must be noted that the surface fluxes over 40°–60°N slowly decreased in all periods (Supplementary Fig. 9d), indicating that they played a minor role in warming. Consistent with previous studies, our results suggest that NA warming may be partly attributed to enhanced AMOC by IO warming[2,17].

The IO warming caused by NA warming may also be attributed to oceanic processes. The ocean heat advection from the western Pacific to the eastern IO through the Indonesian Throughflow (ITF) enhanced the warming effect over the IO. To verify this hypothesis, a mixed layer heat budget analysis for each advection term should be conducted. This issue will be studied in further studies.

We discussed the Pacific effect on the NA-IO warming chain, which could shed light on systematical global interaction that the NA, the IO, and the Pacific are all connected through the change of the Walker Circulation (Supplementary Fig. S3). We will consider the issue of the global chain associated with NA-IO-Pacific in further study.

Questions regarding the weakening and decay of the IO-NA warming chain, which could be broken by a change in ocean circulation over the NA region (e.g., the AMOC), remain unanswered. If the AMO phase changes to negative, NA SST trends become negative and cannot contribute to further IO warming; this change weakens or breaks the warming chain. The role of the Pacific in the IO-NA warming chain may be important for exploring the associated mechanisms. NA warming induces relatively more warming in the western Pacific than in the eastern Pacific, contributing to enhanced ocean heat advection. Fixing the Pacific SST in the PI_NA experiment significantly reduced the IO warming caused by NA SST forcings, suggesting that Pacific behavior is a medium for heat transport from the NA to the IO.

## Methods
**Diagnosis of the observed data**. To obtain the monthly mean SST, we used the National Oceanic and Atmospheric Administration Extended Reconstructed SST version 5[32]. Ocean temperature and thermocline depth data were accessed from the European Centre for Medium-Range Weather Forecasts Ocean reanalysis, and ocean heat content datasets were used for further analysis[33]. Wind and precipitation data were extracted from the National Centers for Environmental Prediction dataset from the NOAA-CIRES Twentieth Century Reanalysis v3 (1943–1992)[34].

**Earth system model**. We used the third version of the Nanjing University of Information Science and Technology Earth System Model (NESM3.0)[30,31,35–37], which consists of atmosphere, ocean, sea ice, and land models that are fully coupled by an explicit coupler. The resolution of the atmospheric model was T63L47. The ocean model has a grid resolution of 1°, with a meridional resolution refined to 1/3° over the equatorial region. The model uses 46 vertical layers, with the upper 15 layers being in the top 100 m. NESM3.0 simulated reasonable climatology with the key characteristics of multi-decadal variabilities.

**Preindustrial, historical, and quadratic $CO_2$ simulations**. We conducted idealized simulations using pre-industrial forcings. Only the ocean model of NESM3.0 was integrated for 4000 years with atmospheric forcings to elucidate the stable upper- and deep-ocean initial conditions. Second, NESM3.0 was integrated for 500 years, with an initial condition that 4000 years of ocean simulation would be performed with preindustrial forcings based on the CMIP6 protocol, to obtain a stable equilibrium state between the atmosphere and ocean. To investigate the warming relationship between the IO and NA regions, we conducted 100-year simulations by nudging the observed SST anomalies over the IO (PI_IO) and increased the SST anomalies uniformly by 1 °C (PI_IO + 1), 2 °C (PI_IO + 2), and 3 °C (PI_IO + 3). Similarly, we repeated the PI experiments with SST anomalies reduced by 1 °C (PI_IO-1), 2 °C (PI_IO-2), and 3 °C (PI_IO-2). Additionally, we conducted the same PI experiments, except for nudged SST, over the NA region (PI_NA and PI_NA±1–3). For historical runs, we conducted four groups of historical simulations. The first was a historical simulation with the observed IO SST (black line in Supplementary Fig. 8a) or climatological SST. The model was freely coupled in other areas, and external forcings were based on the CMIP6 historical simulation protocol. The second was a historical simulation that nudged the observed SST anomalies over the NA region (black line in Supplementary Fig. 8b) or climatological SST. Four ensemble members were used in two groups of the historical simulations. The third was a suite of historical simulations that nudged the observed NA-SST pattern with different trends. We obtained SST data with monotonically reduced trends (0.01–1.02 C 50 years$^{-1}$) by multiplying various scale factors (starting with 0.01 and an increment of 0.05) on the observed NA-SST time series (color lines in Supplementary Fig. 8a). The last was a suite of historical simulations that nudged IO-SST patterns with different trends (0.01–0.7 C 50 years$^{-1}$) by multiplying various scale factors (starting from 0.01 with 0.014 increments) on the observed SST time series over the NA region (color lines in Supplementary Fig. 8b). For the trends of IO (or NA) SST, the very small includes the trends of 0.01–0.25 (°C 70 years$^{-1}$), the small is includes 0.26-0.50, and the large includes 0.51-0.75, the very large includes 0.76-1.02. For HIS_IO, the very small includes the trends of 0.01-0.17 (°C 70 years$^{-1}$), the small includes 0.18–0.35, the large includes 0.36-0.53, and the very large includes 0.57-0.70. The black circle shows observed SST trends.

## Data availability

All observed data used in this study are publicly available (https://psl.noaa.gov/data/gridded/data.20thC_ReanV3.html; https://psl.noaa.gov/data/gridded/data.noaa.ersst.v5.html). The data can be downloaded from https://figshare.com/articles/dataset/IO-NA_warming_chain/19771102.

## Code availability

The codes used in this study can be downloaded here: https://figshare.com/articles/dataset/IO-NA_warming_chain_codes/19771150.

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

## Acknowledgements

Y.-M.Y. is supported by the National Natural Science Foundation of China (NSFC042088101) and the National Research Foundation of Korea (NRF-2022R1A2C1013296). S.-I.A. is supported by the National Research Foundation of Korea (NRF-2018R1A5A1024958). J.-H.P. is supported by the National Research Foundation of Korea (NRF-2020R1C1C1006569). This is Publication No. 379 of the Earth System Modeling Center.

## Author contributions

Y.-M.Y., S.-I.A., and B.W. conceived the idea. Y.-M.Y. performed the model experiments and analyses. S.-I.A., Y.-M.Y., S.-W.Y., B.W, Z.Z., J.L., J.-Y.L., F.L., and J.-H.P. wrote the manuscript. All authors provided critical feedback and helped shape the research, analysis, and manuscript.

## Competing interests

The authors declare no competing interests.
