## [Peer Review File · Nature Communications]

Increased Indian Ocean-North Atlantic Ocean warming chain under greenhouse warmingReviewer #1 (Remarks to the Author):

Yang et al. used coupled model simulations to demonstrate a possible, positive feedback between Indian Ocean warming and North Atlantic warming. Both ocean regions have been persistently warming over the past few decades and have attracted a lot of attention given their global impacts. Previous studies separately found that those two ocean regions may influence each other, and here the authors took a step further by showing their mutual interaction. In general, the manuscript is clearly written, the figures are nicely presented, and the results are comprehensive and reasonable. Several aspects, especially the physical mechanisms underlying their interaction, still need some major revisions. After those are addressed, I think the manuscript can potentially be an interesting contribution to Nature Communications.

Timescale:

It is not clearly described in Methods how long each simulation is and which period is used for analysis. This is critical because timescale matters. The observed climate trends are transient climate changes and are expected to be different from model equilibrium results. I suggest the authors carefully discuss this issue.

Indian Ocean -> North Atlantic:

What is the state of Atlantic Meridional Overturning Circulation (AMOC) in your Indian Ocean warming simulations? Two previous studies using different climate models consistently found that Indian Ocean warming can accelerate the AMOC (Hu and Fedorov 2019; Ferster et al. 2021). Do you see similar results? If so, does the stronger AMOC partly explain the North Atlantic warming? In terms of North Atlantic SST response to Indian Ocean warming, does it have timescale dependence as suggested in Hu and Fedorov (2020)?

North Atlantic -> Indian Ocean:

I understand the mechanism proposed here that the enhanced Pacific-Indian sea level gradient can strengthen the Indonesian Throughflow and potentially warm the Indian Ocean. However, one caveat is that the Indonesian Throughflow is at depth and it is generally hard for the subsurface warming to reach the surface, because upwelling only occurs in very limited areas in the Indian Ocean. See an example in Song et al. (2014).

Indian Ocean warming versus relative Indian Ocean warming:

In the tropics, the rainfall change pattern is mainly affected by the relative SST (local SST minus tropical mean SST) change pattern. Therefore, the relative Indian Ocean warming is probably more relevant than absolute Indian Ocean warming in terms of its teleconnection impacts. I think this point needs to be discussed when you mention the link from Indian Ocean to North Atlantic. Also, how will Fig. 1e be different if relative Indian Ocean SST is used to calculate lead-lag correlation (with long-term trend removed for both series)?

L64-66: The descriptions to those two curves seem to be swapped.

L70-73: The positive correlation comes mainly from the long-term trend. To demonstrate they are correlated on multidecadal timescales, you will firstly need to high-pass filter both series to remove the long-term trend. Their phases do not seem to match well in the 1940s-1970s.

L77: Why did you choose the period after 1970? The post-1970 period is characterized by a continuous long-term warming over both IO and NA without much decadal variability. What you see in the regression patterns (Fig. 1b,c) are mostly due to the common global warming trend, but not the multidecadal variability.

L105: How many ensemble members does each set of experiments have? What relaxation timescale do you use when restoring to the observed SST?

L112: I am curious how well does the free-running historical simulation, without any nudging, reproduce the observed warming over IO and NA. Maybe worthwhile to show that in Figs. 2a and 2b.

Fig. 2c,d: What are the definitions of very small, small, large, and very large?

Fig. 2c: If you nudge the NA SST to observations, why is there still a large inter-member spread in NA SST trend ($0-1^{\circ}\text{C}/50$ years). Shouldn't they be all close to the observed value (black dot)? The same issue for Fig. 2d.

Fig. 4 appears before Fig. 3 in the main text.

L155-156: I understand your point that NA warming without the historical radiative forcing can warm the IO. But the statement here seems a bit odd because the NA warming you impose to the model result partly from global warming. Therefore, the global warming effects have not been entirely excluded here. You can simply say that the NA warming effects on IO also exists when the historical radiative forcing is absent in the model.

L190-200: These results seem out of place here. This study is mostly focused on the connection between IO and NA, and the underlying mechanisms. Their teleconnection impacts on other regions are interesting topics but maybe better saved for future studies.

Fig. 4b-f: Please state clearly what positive values mean – upward or downward?

References:

Ferster, B. S., Fedorov, A. V., Mignot, J., & Guilyardi, E. (2021). Sensitivity of the Atlantic meridional overturning circulation and climate to tropical Indian Ocean warming. *Climate Dynamics*, 1-19.

Hu, S., & Fedorov, A. V. (2019). Indian Ocean warming can strengthen the Atlantic meridional overturning circulation. *Nature climate change*, 9(10), 747-751.

Hu, S., & Fedorov, A. V. (2020). Indian Ocean warming as a driver of the North Atlantic warming hole. *Nature communications*, 11(1), 1-11.

Song, Q., Gordon, A. L., & Visbeck, M. (2004). Spreading of the Indonesian throughflow in the Indian Ocean. *Journal of Physical Oceanography*, 34(4), 772-792.

Reviewer #2 (Remarks to the Author):

This study investigates the connection between the Indian Ocean (IO) and the North Atlantic Ocean (NAO), using a concept of chain to explain the warming trend in both oceans under the global greenhouse warming. The reviewer thinks the authors should give more evidence about the chain that links both oceans. Specific comments are as follows.

A conclusion of this study suggests that the enhanced contrast in the sea levels of the Pacific and IO intensifies Indonesian ThroughFlow (ITF), which increases ocean heat transport from the Pacific and warms the IO (Summary, Line 20-22). Also, the authors suggest that Agulhas current affects the heat transport between the IO and NAO (Line 221-238 and Figure 5). But they do not give sufficient evidence in the main text, but put relevant figures in the supplementary information (Figure S2 and S3). On the other hand, the authors describe the effect of IO-NAO warming chain on the North American precipitation with a short section (Line 189-200 and Figure 3). This part seems independent from the main idea of this study. My major comment is that the authors should focus on the IO-NAO warming chain and explain the mechanism of this concept thoroughly. The impact of the IO-NAO warming may be addressed after the mechanism.

Regarding the mechanism of the role of oceanic advection, the Agulhas current does not directly flow into the southern Atlantic but is in the form of the Agulhas Ring, which features the mesoscale to sub-mesoscale eddies. That also means the schematic (Figure 5) should be modified.

As a chain, all the elements should be linked. After the heat is conveyed into the southern Atlantic, why does not the southern Atlantic warm up, but the NAO gets warmer? The role of the southern Atlantic should be clarified.

Also, the response speed of the atmosphere is faster than the ocean. It is plausible that the warmer NAO induces surface flux anomalies in the IO and wind anomalies in the Pacific. But it seems the ITF and Agulhas current cannot quickly feedback to the NAO warming, not to mention the transport over basin width of the IO. The chain is difficult to accomplish simultaneously, while the author wrote that the maximum correlation coefficient was recorded at zero lag (Line 69).

There are some minor issues that should be corrected.

1. Line 68. Fig. 1f, there is no Figure 1f.

2. Line 149 and 158. Why does Fig. 4a come up earlier than Fig. 3a? As mentioned above, the reviewer suggests a modification of the structure of the main text and figures.

3. The mixed-layer heat budget equation can be put in the method section.

RESPONSE TO REVIEWER COMMENTS

Reviewer #1 (Remarks to the Author):

Yang et al. used coupled model simulations to demonstrate possible, positive feedback between Indian Ocean warming and North Atlantic warming. Both ocean regions have been persistently warming over the past few decades and have attracted a lot of attention given their global impacts. Previous studies separately found that those two ocean regions may influence each other, and here the authors took a step further by showing their mutual interaction. In general, the manuscript is clearly written, the figures are nicely presented, and the results are comprehensive and reasonable. Several aspects, especially the physical mechanisms underlying their interaction, still need some major revisions. After those are addressed, I think the manuscript can potentially be an interesting contribution to Nature Communications.

Re: Thank you for your valuable comments and suggestions. In the revised manuscript, we thoroughly revised the physical mechanism for IO-NA interaction.

1) The impact of the North Atlantic Ocean on the Indian ocean is mainly due to enhanced surface heat fluxes via atmospheric teleconnection. The Indonesian Throughflow (ITF) plays a minor role in IO warming due to weak upwelling in the tropical Indian ocean.

2) The warming of the North Atlantic Ocean by Indian ocean warming mainly results from enhanced surface fluxes by increased longwave radiative heat flux during the initial phase, whereas by enhanced Atlantic Meridional Overturning Circulation (AMOC) during the final phase.

3) We further extended the observed data period from 1950-2020 to consider both global warming and multidecadal variability (Figure 1 in the revised manuscript).

4) We revised the description of model experiments (pre-industrial, historical, and 4CO₂) more clearly.

We believe that the revised manuscript provides better figures and more precise and clear physical mechanisms for IO-NA interaction by addressing your constructive comments.

Timescale:

It is not clearly described in Methods how long each simulation is and which period is used for analysis. This is critical because timescale matters. The observed climate trends are transient climate changes and are expected to be different from model equilibrium results. I suggest the authors carefully discuss this issue.

Re: Thank you for the constructive comment. We appreciate it. In this study, all pre-industrial (PI) and 4CO₂ simulations are integrated for 100 years, of which period is used for analysis. The 100-year seems to represent the transient climate changes, and the equilibrium state by IO or NA SST forcings will be

achieved afterward.

To understand the transient climate change further (herein, only NA response to IO warming is mainly discussed), we examined the temporal evolution of surface heat fluxes over NA and AMOC from PI simulation with IO SST forcings. Results show that surface heat fluxes correspond to the impact of IO on NA through atmospheric teleconnection with short timescales (less than interdecadal), while AMOC shows impact through oceanic circulation with long timescales (greater than multidecadal). Figure R1a shows that NA-SST by IO warming increased rapidly for approximately the initial 20 years and its increasing trends look stagnant by 50-year. For 50-100 years, the NA-SST increases again. The surface net heat flux shows positive for all periods (Fig. R1b); it adjusts very quickly within a few years and then seems to decrease very slowly with time. The longwave radiative flux over NA increases during the initial 20 years and then decreases slowly (Fig. R1c), which is consistent with that of net surface heat flux. This suggests that net surface heat flux plays a major role in the temporal evolution of the SST during the earlier period.

On the other hand, AMOC shows no significant changes until 40-year and then gradually increase up to 100-year (Fig. R1d). The increased AMOC is attributed to enhanced sea surface salinity over the tropical Atlantic Ocean by atmospheric teleconnection (Fig. R1e). It indicates that the IO warming enhances the AMOC with multi-decadal timescales, which probably contributes to NA-SST warming for 50-100 year. These results imply that 100-year simulation reflects “transient climate changes” affected by both atmospheric teleconnection and ocean circulations with different timescales.

These expressions were added to the revised manuscript.

Figure R1 North Atlantic response to Indian Ocean warming. **a** Temporal evolution in annual-mean SST anomalies in “PI_IO+1C”, compared to “PI_IO+0C” model experiments. **b** Same as **a** but for longwave radiative flux. **c** Same as **a** but for net surface heat fluxes. For fluxes, the downward direction is positive (surface warming). The blue (or red) line shows annual mean data and the black line represents an 11-year running mean of the red line (or blue line). **d** Same as **a** but for Atlantic meridional overturning circulation (AMOC) strength. The AMOC strength is estimated as the maximum stream function within 500–5500 m, 30° N to 70° N. The red line shows annual mean data, and the black line shows an 11- year running mean of the red line. **e** Horizontal pattern of sea surface salinity (PSU) anomalies in “PI_IO+1C”, compared to “PI_IO+0C” model experiments.

Indian Ocean -> North Atlantic:

What is the state of Atlantic Meridional Overturning Circulation (AMOC) in your Indian Ocean warming

simulations? Two previous studies using different climate models consistently found that Indian Ocean warming can accelerate the AMOC (Hu and Fedorov 2019; Ferster et al. 2021). Do you see similar results? If so, does the stronger AMOC partly explain the North Atlantic warming? In terms of North Atlantic SST response to Indian Ocean warming, does it have timescale dependence as suggested in Hu and Fedorov (2020)?

Re: In our simulations, the AMOC is significantly enhanced by IO warming through atmospheric teleconnection. In the initial phase, the AMOC shows no significant changes but it increases continuously for 40-100 years (Fig. R1d). This temporal evolution of AMOC is consistent with previous studies (Hu and Fedorov 2019; Ferster et al. 2021). Compared to the SST trend over NA, the AMOC could partly contribute to NA warming during the final phase

The NA-SST increases rapidly during the initial 20 years and then stagnant by 50-year and slightly increases again up to 100 years (Fig. R1a). The trends of surface fluxes are slightly reduced all period, suggesting that the surface fluxes contribute to IO warming more during the initial period. The AMOC show significant increasing trends during 40-100 years, indicating that the AMOC partly explain the NA warming during later periods.

We examined the timescale dependency of high-latitude (50°-70°N) NA-SST by IO warming. Figure R2a and 2b show the horizontal pattern of SST anomalies between “PI_IO+1C” and “PI_IO+0C” experiments during the initial phase (0-30years) and final phase (70-100 years), respectively. In the initial phase, the moderate warming over equatorial and subtropical Atlantic region with relatively weak warming near 40°-60°N. The weak warming in the south of Greenland changed to basin-wide strong warming there during the final phase (see Fig. R2b; Fig. R2c). These results suggest that the NA warming may be partly attributed to enhanced AMOC by IO warming and our results are consistent with Hu and Fedorov (2020). The enhanced AMOC by the Indian Ocean warming could increase northward transport of ocean heat and then the whole North Atlantic warms by the strengthened AMOC (Fig. R1d). These discussions and related figures were added to the revised manuscript.

Figure R2 North Atlantic response to Indian ocean warming. **a** Horizontal patterns of SST and surface wind anomalies between PI_IO+1C and PI_IO + 0C model experiments during the initial phase (0-30yrs). **b** Same as a but for the final phase (70-100yrs). **c** temporal evolution of SST anomalies averaged over 0°-80°W, 50°-70°N between PI_IO+1C and PI_IO + 0C model experiments. **d** Same as c but for net surface fluxes ($W m^{-2}$). “Initial” refers to an average for Years 1–30, while “final” refers to Years 70–100. The blue (or red) line shows annual mean data and the black line represents an 11-year running mean of the red line (or blue line)

North Atlantic -> Indian Ocean:

I understand the mechanism proposed here that the enhanced Pacific-Indian sea level gradient can strengthen the Indonesian Throughflow and potentially warm the Indian Ocean. However, one caveat is that the Indonesian Throughflow is at depth and it is generally hard for the subsurface warming to reach the surface because upwelling only occurs in very limited areas in the Indian Ocean. See an example in Song et al. (2014).

Re: Thanks for the constructive comments. Based on the reviewer's comment, we carefully examined the SST, surface heat fluxes, surface wind, and ocean upwelling in the model with NA warming forcing. The NA warming induces basin-wide SST warming over tropical Indian ocean between 30°S and 30°N (Fig. R3a). We examined latent heat fluxes and surface wind anomalies. By NA warming, the surface

winds are weakened and then latent heat fluxes are reduced by atmospheric teleconnection (Fig. R3b), contributing to IO warming significantly (Fig. R3a). We additionally explore ocean upward water fluxes over the IO region. When NA warming is given in the model, the ocean upward fluxes increase in the north Indian ocean region, while they decrease over the south Indian ocean (0° - 30° S) where Indonesian throughflows (ITF) are advected from western Pacific (Fig. R3c). ITF is advected through the subsurface and so the strong upwelling should be accompanied to warm the surface. However, the upwelling is weak or negative over the equatorial or south of the Indian ocean, suggesting that ITF plays a minor role in IO warming. These results are consistent with previous studies (Li et al. 2015, Song et al. 2004).

Figure R3 Indian ocean response to north Atlantic ocean warming. a Horizontal patterns of SST anomalies ($^{\circ}$ C) between PI_NA+1C and PI_NA+0C model experiments during total periods (0-100yrs). **b** Same as a but for latent heat fluxes (W m^{-2}) and surface winds (m s^{-1}). **c** same as but for vertical advection by upwelling between 50-100m or of ocean model ($\text{X}10 \text{ g m}^{-2} \text{ s}^{-1}$). 100year simulated data are used for analysis.

Indian Ocean warming versus relative Indian Ocean warming:

In the tropics, the rainfall change pattern is mainly affected by the relative SST (local SST minus tropical mean SST) change pattern. Therefore, the relative Indian Ocean warming is probably more relevant than absolute Indian Ocean warming in terms of its teleconnection impacts. I think this point needs to be discussed when you mention the link from the Indian Ocean to North Atlantic. Also, how will Fig. 1e be different if relative Indian Ocean SST is used to calculate lead-lag correlation (with long-term trend removed for both series)?

Re: We agree with the reviewer's comments. We examined the relationship between IO and NA SST using observed data (1950-2020, detrended). The peak of lead-lag correlation between local IO and NA SST occurs at 0-years with a coefficient of 0.8 to 0.9 (Fig. R4a). The correlation decreases with increase

lag (lead) years and shows a correlation coefficient of 0.7 at -8years lag and 0.5 at +10 years lag. On the other hand, we examined the lead-lagged correlation between relative Indian ocean (RIO) and North Atlantic (NA) SST, as the reviewer mentioned. The peak correlation occurs 0-2yrs and second peaks are found -8yrs (it means IO lead NA with 8yr lags) and +10yrs (NA lead IO with 10yr lags) (Fig. R4b). Compared to those with IO and NA SST, the correlation patterns between RIO and NA are similar but their magnitudes are weakened.

Figure R4 Relationship between relative IO and NA **a.** Observed lead-lag correlation coefficient between the IO and NA indices during 1950-2020. The lag is positive (negative) when the IO leads (lags). The black line represents SST data that are removed long-term linear trends and the red line shows original SST data. **b.** same as a but for relative IO and NA. The relative IO and NA are defined as average SST in the Indian Ocean (30°S–30°N, 40°E–120°E) minus the whole tropical ocean (30°S–30°N, 0°-360°E).

L64-66: The descriptions of those two curves seem to be swapped.

Re: We corrected them in the revised manuscript. Figure 1 in the old manuscript was modified using detrended observed SST to remove anthropogenic warming.

“To quantify the strength of the IO-NA warming chain, we defined the IO SST index (IO index) and the NA Ocean SST index (NA index) by using the SST averaged over each box (Fig. R5). Noted that the long-term linear trends of the SST were removed. From 1970–2020, the NA index showed a warming trend with 0.12 °C per decade, with strong multi-decadal fluctuations (Fig. 1d, red line). Local maximum and minimum NA indices were observed close to 1940 and 1970, respectively. The IO index also

showed a significant warming trend (0.06 °C per decade) from 1950 to 2020 and a maximum interdecadal fluctuation close to 1940. The warming trends and interdecadal-to-multidecadal fluctuations in the IO and NA indices tend to overlap”.

Figure R5. Historical warming trends. Observed time series of the NA (red line) and IO (blue line) indices. Long-term linear trends in the SST data were removed before analysis and the last 5 years were excluded from the analysis.

L70-73: The positive correlation comes mainly from the long-term trend. To demonstrate they are correlated on multidecadal timescales, you will firstly need to high-pass filter both series to remove the long-term trend. Their phases do not seem to match well in the 1940s-1970s.

Re: We removed the long-term trends of the total SST by applying the high-pass filter (50 year). As shown in FigR4, the lead-lag correlation between NA and IO using the detrended SST data is relatively lower than those with the trend. When we use a longer period of SST data (1900-2020), the maximum correlation of SST data without detrending between NA and IO is 0.85 (Fig. R6), while the correlation with detrending SST data is reduced with a correlation of 0.61. These results suggest that IO and NA are positively correlated and the relationship could be intensified by the global warming effect.

Figure R6 Observed lead-lag correlation coefficient between the IO and NA indices during 1900-2020. The lag is positive (negative) when the IO leads (lags). The black line represents SST data that are

removed long-term linear trends and the red line shows original SST data.

L77: Why did you choose the period after 1970? The post-1970 period is characterized by continuous long-term warming over both IO and NA without much decadal variability. What you see in the regression patterns (Fig. 1b,c) are mostly due to the common global warming trend, but not the multidecadal variability.

Re: As the reviewer mentioned, we increase the data period from 1950-2020 to consider multidecadal variability and modified Figure 1b-c in the original manuscript (Fig. R7 below is the same) with detrended SST data. Fig. R7a global SST anomalies and surface winds regressed onto the NA index by using observational data from 1950 to 2020. The NA warming pattern resembles a developing positive phase of the Atlantic Multidecadal Oscillation (AMO), and strong warming is observed in the tropical IO, northwestern and southern Pacific. When we calculated the same regressed pattern using detrended SST data (Fig. R7b), there are changes in the other two ocean basins. The warming in the IO is weakened but still significant. The northeastern Pacific cooling is strengthened and the warming in the north of the Pacific extended eastward. Finally. The pattern of SST anomalies in the Pacific is similar to the negative phase of the inter-decadal Pacific Oscillation (Fig. R7b).

Second, we examined global SST anomalies and surface winds regressed onto the IO index by using observational data from 1950 to 2020. Strong warmings are seen in the north Atlantic ocean, western and eastern Pacific, and south of Pacific but weak cooling in the northern Pacific (Fig. R7c). When we calculated the same regressed pattern using detrended SST data, NA warming is significantly intensified. The warming in the Pacific is strengthened (Fig. R7d). These results indicate that the NA and IO are positively correlated with each other with a multidecadal timescale.

Figure R7. a Observed SST (K) and surface wind anomalies (arrow, m/s) regressed onto the NA index (0° – 70° N, 80° – 0° W). The 11-year running average data for 1950–2020 were used for the analysis. The hatched area represents the regressed SSTA and is significant at the 95% confidence level. **b** Same as **a** but for detrended SST. **c** Same as **a** but for the IO index (30° S– 30° N, 40° – 120° E), **d** same as **b** but the IO index (30° S– 30° N, 40° – 120° E).

L105: How many ensemble members does each set of experiments have? What relaxation timescale do you use when restoring to the observed SST?

Re: The ensemble number is four for each historical run with the observed NA-SST (or IO-SST) trends. The relaxation time scale for the observed SST nudging is 1.5 days. The external forcings were based on historical simulations suggested by the CMIP6 project. The models were integrated for 1900-2015.

L112: I am curious how well does the free-running historical simulation, without any nudging, reproduces the observed warming over IO and NA. Maybe worthwhile to show that in Figs. 2a and 2b.

Re: As the reviewer mentioned, we examined the regressed SST and surface wind on NA from historical simulation without any nudging. Figure R8 shows the model can capture the observed SST pattern. The model simulates IO warming related to NA warming with a slightly strong magnitude. The model also captures the strong warming in the northern and southern Pacific. Corresponding surface wind anomalies in the Indian Ocean and Pacific are similar with observation. Figure R8d shows regressed SST and surface wind on IO from historical simulation. The model simulates NA warming associated with IO warming with slightly stronger amplitude and captures warming in the southern and eastern Pacific. Note that it fails to capture cooling in the northern Pacific. In summary, the model captures the observed IO-NA warming pattern reasonably well. These descriptions for the performance of the model were added in the revised manuscript.

Figure R8. **a** Observed SST (K) and surface wind anomalies (arrow, m s^{-1}) regressed onto the NA index (0° – 70°N , 80° – 0°W). The hatched area represents the regressed SSTA and is significant at the 95% confidence level. **b** Same as **a** but from model simulated SST. **c** Same as **a** but for the IO index (30°S – 30°N , 40° – 120°E), **d** same as **b** but the IO index (30°S – 30°N , 40° – 120°E). The model simulation is conducted by free historical forcings. The 11-year running average data were used for 1950–2020.

Fig. 2c,d: What are the definitions of very small, small, large, and very large?

Re: We classified the historical experiment based on NA (or IO) warming trends. For the historical run with NA SST nudging, “the very small” includes the trends of 0.01-0.25 ($\text{C } 50 \text{ yrs}^{-1}$), “the small” is 0.26-0.50, “the large” is 0.51-0.75, “the very large” is 0.76-1.02. For the run with IO SST nudging, “the very small” includes the trends of 0.01-0.17 ($\text{C } 70 \text{ yrs}^{-1}$), “the small” is 0.18-0.35, “the large” is 0.36-0.53, “the very large” is 0.57-0.70.

Fig. 2c: If you nudge the NA SST to observations, why is there still a large inter-member spread in the NA SST trend ($0\text{-}1^{\circ}\text{C}/50 \text{ years}$). Shouldn't they be all close to the observed value (black dot)? The same issue is for Fig. 2d.

Re: The explanation for historical simulation was not clear. We conducted four groups of historical simulations. First is a historical simulation with the observed IO SST (the black line in Fig. R9a). The model was freely coupled in other areas, and external forcings were based on the CMIP6 historical simulation protocol. The second is a historical simulation by nudging observed SST anomalies over the NA region (black line in Fig. R9b).

The third is a suite of historical simulations by nudging the observed NA-SST pattern with different

trends. We obtained SST data with monotonically reduced trends ($0.01\text{--}1.02\text{ C } 50\text{ yrs}^{-1}$) by multiplying various scale factors (starts from 0.01 with 0.05 increments) on the observed NA-SST time series (color lines in Fig. R9a). The last is a suite of historical simulations by nudging observed IO-SST patterns with different trends ($0.01\text{--}0.7\text{ C } 50\text{ yrs}^{-1}$) by multiplying various scale factors (starts from 0.01 with 0.014 increments) on the observed SST time series over the NA region (color lines in Fig. R9b). These descriptions were added to the revised manuscript.

Figure R9. **a** Time series of annual mean observed IO SST. Black line shows observed SST anomalies and color lines represent IO SST with different trends. The trend of the black line is 0.0145C year^{-1} and the trends of each color line are monotonically reduced (interval is 0.05) and the trend of the bottom line is 0.01. **b** Same as a but for NA SST.

Fig. 4 appears before Fig. 3 in the main text.

Re: We corrected the revised manuscript. We explained the results of the PI simulation first and then historical and 4CO_2 experiments in the revised manuscript.

L155-156: I understand your point that NA warming without the historical radiative forcing can warm the IO. But the statement here seems a bit odd because the NA warming you impose to the model result partly from global warming. Therefore, the global warming effects have not been entirely excluded here. You can simply say that the NA warming effects on IO also exists when the historical radiative forcing is absent in the model.

Re: We agreed with the reviewer's comment. We modified those expressions in the revised manuscript—"The above-mentioned simulation results suggest that the NA warming effects on IO also exist when historical radiative forcings are absent in the model".

L190-200: These results seem out of place here. This study is mostly focused on the connection between IO and NA, and the underlying mechanisms. Their teleconnection impacts on other regions are interesting topics but maybe better saved for future studies.

Re: We removed the impact of IO-NA warming on precipitation in the U.S. and shortly mentioned it as a topic for further studies in the discussion- "It may be interesting topics to explore the impact of IO-NA interaction on midlatitudes (e.g. precipitation, temperature and fire emission and so on) in the further studies."

Fig. 4b-f: Please state clearly what positive values mean – upward or downward?

Re: In this manuscript, positive means downward and represent surface warming by each surface fluxes. We added those expressions in the caption of the figure.

References:

Ferster, B. S., Fedorov, A. V., Mignot, J., & Guilyardi, E. (2021). Sensitivity of the Atlantic meridional overturning circulation and climate to tropical Indian Ocean warming. *Climate Dynamics*, 1-19.

Hu, S., & Fedorov, A. V. (2019). Indian Ocean warming can strengthen the Atlantic meridional overturning circulation. *Nature climate change*, 9(10), 747-751.

Hu, S., & Fedorov, A. V. (2020). Indian Ocean warming as a driver of the North Atlantic warming hole. *Nature communications*, 11(1), 1-11.

Song, Q., , A. L., & Visbeck, M. (2004). Spreading of the Indonesian throughflow in the Indian Ocean. *Journal of Physical Oceanography*, 34(4), 772-792.

Li, X., Xie, SP., Gille, S. et al. Atlantic-induced pan-tropical climate change over the past three decades. *Nature Clim Change* 6, 275–279 (2016).

Reviewer #2 (Remarks to the Author):

This study investigates the connection between the Indian Ocean (IO) and the North Atlantic Ocean (NAO), using a concept of chain to explain the warming trend in both oceans under the global greenhouse warming. The reviewer thinks the authors should give more evidence about the chain that links both oceans. Specific comments are as follows.

A conclusion of this study suggests that the enhanced contrast in the sea levels of the Pacific and IO intensifies Indonesian ThroughFlow (ITF), which increases ocean heat transport from the Pacific and warms the IO (Summary, Line 20-22). Also, the authors suggest that Agulhas current affects the heat transport between the IO and NAO (Line 221-238 and Figure 5). But they do not give sufficient evidence in the main text, but put relevant figures in the supplementary information (Figure S2 and S3). On the other hand, the authors describe the effect of IO-NAO warming chain on the North American precipitation with a short section (Line 189-200 and Figure 3). This part seems independent from the main idea of this study. My major comment is that the authors should focus on the IO-NAO warming chain and explain the mechanism of this concept thoroughly. The impact of the IO-NAO warming may be addressed after the mechanism.

Regarding the mechanism of the role of oceanic advection, the Agulhas current does not directly flow into the southern Atlantic but is in the form of the Agulhas Ring, which features the mesoscale to sub-mesoscale eddies. That also means the schematic (Figure 5) should be modified.

As a chain, all the elements should be linked. After the heat is conveyed into the southern Atlantic, why does not the southern Atlantic warm up, but the NAO gets warmer? The role of the southern Atlantic should be clarified.

Also, the response speed of the atmosphere is faster than the ocean. It is plausible that the warmer NAO induces surface flux anomalies in the IO and wind anomalies in the Pacific. But it seems the ITF and Agulhas current cannot quickly feedback to the NAO warming, not to mention the transport over basin width of the IO. The chain is difficult to accomplish simultaneously, while the author wrote that the maximum correlation coefficient was recorded at zero lag (Line 69).

Re: From the reviewer's comments, we realized that our mechanism for the IO-NA mechanism suggested in the earlier manuscript has a weakness and the associated evidence is not clear. In the revised manuscript, we thoroughly revised the physical mechanism for IO-NA interaction. 1) The impact of the North Atlantic Ocean on the Indian ocean is mainly due to enhanced surface fluxes by atmospheric teleconnection. The Indonesian flow plays a minor role in IO warming because upwelling in the tropical Indian ocean is weak and limited.

2) The warming of the North Atlantic Ocean by Indian ocean warming mainly results from enhanced surface fluxes by increased longwave radiative and latent heat flux through atmospheric teleconnection during the initial phase. During the later phase, NA warming is attributed to enhanced AMOC, which are associated with enhanced salinity over tropical Atlantic regions (Hu and Fedorov 2019). We removed

the explanation for the role of Agulhas current because the model cannot precisely simulate the impact of Agulhas current on NA warming due to the relatively coarse resolution of the ocean model.

3) We removed the impact of IO-NA warming on precipitation in the U.S. in the main text and shortly mentioned it as a topic for further studies in the discussion.

We believe that the revised manuscript provides better figures and more precise and clear physical mechanisms for IO-NA interaction by addressing your constructive comments.

a. Impact of IO warming on NA

Figure R1a shows the observed SST and surface wind anomalies regressed on the IO index during 1950-2020. Similarly, Figure R1b shows the simulated NA-SST response to the 1°C IO-warming forcing reflecting the observed warming pattern. In both the observed and simulated results, strong warmings are seen in the north Atlantic ocean, western and eastern Pacific, and south of Pacific but no significant changes in the northern Pacific. Strong easterlies are seen in the western Pacific and Indian ocean.

Figure R1c shows temporal evolution of NA-SST by IO warming. The NA-SST increased rapidly for 10-20yrs and then its increasing trends are weakened. To investigate the effect of the surface heat flux on NA SST, we conducted a surface heat budget analysis over the NA region. Figure R1d shows the difference between PI_IO+1C and PI_IO+0C. The net surface flux is a significantly positive and steady-state during all periods (Fig. R1d), suggesting that net surface flux is adjusted to the IO-SST forcings. Overall, during the initial periods, the net surface heat flux may play a major role in the NA-SST warming.

On the other hand, we further examined changes in AMOC (Fig. R1e), which could affect SST warming by meridional transport of ocean heat. The AMOC shows no significant changes for first 40 years and then gradually increase up to 100yrs, indicating that the IO warming slowly enhances the AMOC strength with a longer time scale. The enhanced AMOC may contribute to further increased NA-SST during the later periods (Fig. R1c). The ascending motion in the IO generates a descending motion in the tropical Atlantic region (e.g. Supplementary Fig. S1b), inducing sea surface salinity in the tropical Atlantic region (Fig. R1f). The higher sea surface salinity (Fig. R1f), of which transportation northward contributes to remain net surface heat flux during a couple of decades and partly enhanced AMOC after a few decades.

Figure R1 **a** Observed SST (K) and surface wind anomalies (arrow, m s^{-1}) regressed onto the IO index (0° – 70°N , 80° – 0°W). The 11-year running average data were used for 1950–2020. The hatched area represents the regressed SSTA and is significant at the 95% confidence level. **b** Anomalies in SST ($^{\circ}\text{C}$) and surface wind from the fixed preindustrial (PI) simulation with observed IO SST anomalies. **c** Temporal evolution in annual-mean SST anomalies. **d** Same as **b** but for net surface heat flux (positive values represent downward direction, W m^{-2}). **e** Same as **b** but for Atlantic meridional overturning circulation (AMOC) strength. The AMOC strength is estimated as the maximum stream function within 500–5500 m, 30°N to 70°N . The blue (or red) line shows annual mean data and the black line represents an 11-year running mean of the red line (or blue line). **f** Same as **b** but for sea surface salinity. The anomalies are computed for the 100 years of the PI_NA+1C experiment with respect to the pre-industrial simulation (PI_NA+0C).

b. Impact of NA warming on the IO

In the IO response to the NA warming (difference between PI_NA+1C and PI_NA+0C), the model showed significant IO warming (Fig. R2b). Warming also occurred in the northern and southern subtropical Pacific and equatorial western Pacific (Fig. R2b); however, relatively cool SST prevailed in the equatorial eastern Pacific. The anticyclonic flows in the north and south Pacific generate easterly anomalies in the western Pacific that may increase ocean heat transport from the western Pacific to the eastern Indian Ocean. The IO-SST pattern induced by NA warming from model largely resemble those in the observations.

Fig. R2c shows the difference in surface heat fluxes between PI_NA+1C and PI_NA+0C experiments, wherein positive means downward heat flux to the ocean, thus contributing to SST warming. It is found that the weakened wind by enhanced easterly anomalies is influential most, so IO-SST change appears to be affected by surface heat fluxes (mostly by LH) associated with surface wind change (Fig. R2c).

The anticyclonic flows in the north and south Pacific link to the easterly anomalies in the western equatorial Pacific that may increase ocean heat transport from the western Pacific to the eastern Indian Ocean via ITF. Because the ITF is advected through the subsurface, the strong upwelling should be accompanied to warm the surface. These processes allow us to infer that ocean heat advection from the western Pacific to the eastern IO through the Indonesian throughflow (ITF) enhances SST warming over the IO. However, it is found that the upwelling is weak (negative) over the equatorial (south) Indian ocean, suggesting that ITF plays a minor role in IO warming (Fig. R2d). These results are consistent with previous studies (Li et al. 2015, Song et al. 2004).

Figure R2. **a** Observed SST (K) and surface wind anomalies (arrow, m s^{-1}) regressed onto the NA index (0° – 70°N , 80° – 0°W). The 11-year running average data were used for 1950–2020. The hatched area represents the regressed SSTA and is significant at the 95% confidence level. **b** Anomalies in SST ($^{\circ}\text{C}$) and surface wind (m s^{-1}) response to the NA-SST warming forcing reflecting observed IO-SST warming pattern under PI simulations. **c** Same as b but for latent heat flux and surface wind. Positive color means downward. **d**. Same as b but for upward ocean water fluxes. Positive colors mean upward. The anomalies are computed for the 100 years of the PI_NA+1C experiment with respect to the pre-industrial simulation (PI_NA+0C).

There are some minor issues that should be corrected.

1. Line 68. Fig. 1f, there is no Figure 1f.

Re: It was corrected.

2. Line 149 and 158. Why does Fig. 4a come up earlier than Fig. 3a? As mentioned above, the reviewer suggests a modification of the structure of the main text and figures.

Re: We corrected the revised manuscript. We explained the results of PI simulation first and then historical and 4CO2 experiments in the revised manuscript.

3. The mixed-layer heat budget equation can be put in the method section.

Re: We added the mixed-layer heat budget equation in the “method” section of the revised manuscript-“Heat budget analysis of the ocean mixed layer temperature tendency is used to quantify the contributions of different processes to the Indian ocean warming. This diagnostic equation can be derived as follows:

$$\frac{\partial T'}{\partial t} = -(\mathbf{V}' \cdot \nabla \bar{T} + \bar{\mathbf{V}} \cdot \nabla T' + \mathbf{V}' \cdot \nabla T') + \frac{Q'_{net}}{\rho C_p H} + R = - \left[\left(\frac{u' \partial \bar{T}}{\partial x} + \bar{u} \frac{\partial T'}{\partial x} + \frac{u' \partial T'}{\partial x} \right) + \left(\frac{v' \partial \bar{T}}{\partial y} + \bar{v} \frac{\partial T'}{\partial y} + \frac{v' \partial T'}{\partial y} \right) + \left(\frac{w' \partial \bar{T}}{\partial z} + \bar{w} \frac{\partial T'}{\partial z} + \frac{w' \partial T'}{\partial z} \right) \right] + \frac{Q'_{net}}{\rho C_p H} + R,$$

where T denotes the mixed-layer temperature; $\mathbf{V}=(u, v, w)$ represents the zonal and meridional currents and upwelling velocities, respectively; The mixed-layer depth H is taken as a constant 50 m, which is not sensitive to the different mixed-layer thickness in this study, such as $H = 30$ or 70 m. For a detailed introduction of data and method please refer to “Supplementary Information.”

References

- Hu, S., Fedorov, A.V. Indian Ocean warming can strengthen the Atlantic meridional overturning circulation. *Nat. Clim. Chang.* 9, 747–751 (2019). <https://doi.org/10.1038/s41558-019-0566-x>
- Song, Q., , A. L., & Visbeck, M. (2004). Spreading of the Indonesian throughflow in the Indian Ocean. *Journal of Physical Oceanography*, 34(4), 772-792.
- Li, X., Xie, SP., Gille, S. et al. Atlantic-induced pan-tropical climate change over the past three decades. *Nature Clim Change* 6, 275–279 (2016).

Reviewer #1 (Remarks to the Author):

I think the authors have reasonably addressed most of my concerns and the concerns raised by the other reviewer. I have a few additional minor suggestions.

L27: "Positive interaction" sounds odd. How about "This two-way interaction (i.e., the IO-NA warming chain) acts as a positive feedback that reinforces ..."

L69: "For both SST indices, the long-term trend has been removed and a 11-year running mean has been applied before further analysis is conducted"

L69: Delete "also"

L73: "around 1940 and 1970, respectively"

Fig. 1b,c: Are you sure they are based on detrended data? Look different from Fig. R7 in the response file. Please double check all the figures and captions and make sure they are correct.

L129: Describe here how many years each simulation is conducted and which period is used for averaging.

L142: Fig. 2c?

L149: Delete "However"

L166-172: It is fairly well known that oceanic upwelling is absent in the equatorial Indian Ocean. However, this cannot be used as the direct evidence for the absence of contribution from oceanic processes to the Indian Ocean warming. For a more targeted attribution, you will need to do a mixed layer heat budget analysis for each advection term. If the authors decide to leave it as future work, please add a discussion on this issue with speculations and caveats clearly stated.

L195: Fig. 4a?

L202: Fig. 4b?

L220: Fig. 4c,d?

Please double check the figure references and the order of figures.

Reviewer #2 (Remarks to the Author):

Review to Increased Indian Ocean-North Atlantic Ocean warming chain under greenhouse warming by Yang et al.

General Comments

The authors have made a lot of changes in the revision. The reviewer read this manuscript as a new contribution and found some interesting points. Compared with the last version, this revision removes descriptions on the Agulhas current and precipitation in the U.S, suggests a weaker role of the ITF, and further emphasizes the role of atmospheric forcings like longwave radiative and latent heat flux. The reviewer thinks these modifications are closer to the authors' topic. However, the reviewer still has some concerns regarding the mechanism of the IO-NA Warming Chain in the revised paper. The most important issue is that the evidence of the NA-IO Warming Chain is enough, while the behind mechanism is still unclear. Specific comments are as follows.

Major Comments

Question 1: Role of Pacific

The reviewer found that the role of the Pacific should be important in the Chain, as the authors also mentioned the atmospheric motion (ascending/descending) in the Pacific (Lines 94-127, Fig. S3). Analyses of this part are based on observational data. However, the authors do not give the model results to confirm this view. How are the model results in terms of the role Pacific?

On the other hand, how are the relationships between the Pacific indices (e.g., IPO, PDO, and decadal ENSO index) and IO-NA indices? Since tropical atmospheric motions are driven mainly by SST below, Fig. S3 may also suggest large changes of Pacific SST and the Pacific Walker Circulation. It is plausible that the three major oceans are in the same Warming Chain.

Besides, the negative IPO phase pattern is unclear in Fig. 1a (Lines 98-99) compared to the typical IPO pattern (cf. Henley et al., 2015).

Henley, B.J., Gergis, J., Karoly, D.J. et al. A Tripole Index for the Interdecadal Pacific Oscillation. *Clim Dyn* 45, 3077–3090 (2015).

Question 2: How the NA warming inducing the IO Warming

The authors do not clarify how the NA induces IO warming. In Line 165-167, they suggested that the IO warming caused by NA warming may also be attributed to oceanic processes associated with the ITF. BUT, the last of this paragraph, they wrote that “the ITF played a minor role in IO warming”. I am confused. What is the major factor in this part of the Warming Chain?

Besides, in Fig. 2h, how should we understand “ocean upward water fluxes”? It is shown in cool color. Does that mean the negative effect of upwelling and ITF on IO warming? If so, how is it important for the ITF? Also, please explain where Fig. 2h is for citation.

Moreover, since the correlation analyses (Fig. 1e) suggest the simultaneous coefficient is the largest, does this mean the atmospheric forcing is the major cause? Because the oceanic process needs time to respond (as seen in the AMOC strengthened in Fig. 2e). If so, how does the atmosphere forcing related to the NA warming affect the IO warming?

Question 3: Mechanism verified by model results

In the model results, the authors showed sufficient evidence that there is a Warming Chain between the NA and the IO, in which the changes of SST in both oceans are in a similar linear change (Figs. 3-4). However, physical processes can be further explored using the model results since the observational data are short.

One concern is what leads to the surface flux change in the IO-NA (Fig. 2). For example, the Rossby wave chain or the global Walker Circulation change may contribute to the IO-NA warming. So far, the authors just showed the final results rather than the processes behind them. At least, they can verify Fig. S3 by the model results.

Other minor comments

It seems that there is no content for mixed-layer heat budget analysis in the revision, but it is listed in the method.

Citations for Figure 4 are all wrongly replaced by Figure 2. (See section IO-NA warming chain evidenced through historical simulations)

References should be listed correctly. (See No. 3 and 21, no published year; No. 7 and 16, duplication; other many spelling errors.)

RESPONSE TO REVIEWER COMMENTS

Reviewer #1 (Remarks to the Author):

I think the authors have reasonably addressed most of my concerns and the concerns raised by the other reviewer. I have a few additional minor suggestions.

Re: We thank the reviewer for your contribution to the peer review of this work. We appreciate it. In the revised manuscript, we thoroughly revised the explanation for each experiment, citation for figures, and references. We realized that the ITF hardly contributes to the IO warming, so we removed the role of ITF in the main text and briefly mentioned it in the ‘Discussion’ section. We believe our manuscript was improved a lot by addressing the reviewer’s comments.

L27: “Positive interaction” sounds odd. How about “This two-way interaction (i.e., the IO-NA warming chain) acts as a positive feedback that reinforces ...”

Re: We agreed with the reviewer’s suggestion. We modified those expressions in the revised manuscript-“This two-way interaction (i.e., the IO-NA warming chain) acts as positive feedback that reinforces further warming over both ocean basins.”

L69: “For both SST indices, the long-term trend has been removed and an 11-year running mean has been applied before further analysis is conducted”

RE: We modified that sentence as the reviewer suggested-“For both SST indices, the long-term trend has been removed and an 11-year running mean has been applied before further analysis was conducted”.

L69: Delete “also”

Re: We removed it.

L73: “around 1940 and 1970, respectively”

Re: We corrected it-“Local maximum and minimum NA indices were observed around 1940 and 1970, respectively.”

Fig. 1b,c: Are you sure they are based on detrended data? Look different from Fig. R7 in the response file. Please double-check all the figures and captions and make sure they are correct.

Re: Figure R1 was corrected in the revised manuscript. Also, we checked all figures and captions and corrected them.

Figure R1: Historical warming trends. **a** Observed trends (1950–2020) in annual-mean sea surface temperature (SST; units: $^{\circ}\text{C } 50 \text{ year}^{-1}$). The red (blue) box represents the region where the Indian (North Atlantic) Ocean index was defined. **b-c** Observed SST (K) and surface wind anomalies (arrow, m s^{-1}) regressed onto **(b)** the North Atlantic Ocean (NA) index (0° – 70°N , 80° – 0°W) and **(c)** Indian Ocean (IO) index (30°S – 30°N , 40° – 120°E). The hatched area represents the regressed SST and is significant at the 95% confidence level. **d** Observed time series of the NA (red line) and IO (blue line) indices. **(e)** The observed lead-lag correlation coefficient between the IO and NA indices. The lag is positive (negative) when the IO leads (lags) and the grey lines represent a 95% significance level. For **b-d**, the 11-year running average data were used for 1950–2020 and long-term linear trends were removed. For **e**, the last 5 years were excluded from the analysis, and the long-term linear trends in the SST data were

removed before regression for the only black line.

L129: Describe here how many years each simulation is conducted and which period is used for averaging.

Re: We added the information of the model simulation period and data period used for the analysis-“ The model in this study was integrated from 1900 to 2020 and 71-year data (1950-2020) are utilized for the analysis. To check the model’s capability on the simulation of the observed IO-NA relationship, the observed regression maps of global SST and surface wind anomalies onto the NA (or IO) for 1950-2020 resemble those simulated, inferring that the model is capable to simulate the observed IO-NA relationship”

L142: Fig. 2c?

Re: It was corrected. We also checked the citations for all figures and corrected them.

L149: Delete “However”

Re: It was deleted.

L166-172: It is fairly well known that oceanic upwelling is absent in the equatorial Indian Ocean. However, this cannot be used as direct evidence for the absence of contribution from oceanic processes to the Indian Ocean warming. For a more targeted attribution, you will need to do a mixed layer heat budget analysis for each advection term. If the authors decide to leave it as future work, please add a discussion on this issue with speculations and caveats clearly stated.

Re: As the reviewer suggested, we think that it would be better to add these parts in the “discussion” rather than the “results” section, which is partly because the oceanic process is slower than the atmospheric process so it is not suitable for explanation simultaneous feedback

process. We added those expressions in the discussion section-“ The IO warming caused by NA warming may also be attributed to oceanic processes. The ocean heat advection from the western Pacific to the eastern IO through the Indonesian Throughflow (ITF) enhanced the warming effect over the IO. To verify this hypothesis, a mixed layer heat budget analysis for each advection term should be conducted. This issue will be studied in the further studies.”

L195: Fig. 4a?

It was corrected

L202: Fig. 4b?

It was corrected

L220: Fig. 4c,d?

It was corrected

Please double-check the figure references and the order of the figures.

Re: We checked the figure references.

Reviewer #2 (Remarks to the Author):

Review to Increased Indian Ocean-North Atlantic Ocean warming chain under greenhouse warming by Yang et al.

General Comments

The authors have made a lot of changes in the revision. The reviewer read this manuscript as a new contribution and found some interesting points. Compared with the last version, this revision removes descriptions on the Agulhas current and precipitation in the U.S, suggests a weaker role of the ITF, and further emphasizes the role of atmospheric forcings like longwave radiative and latent heat flux. The reviewer thinks these modifications are closer to the authors' topic. However, the reviewer still has some concerns regarding the mechanism of the IO-NA Warming Chain in the revised paper. The most important issue is that the evidence of the NA-IO Warming Chain is enough, while the behind mechanism is still unclear. Specific comments are as follows.

Re: We appreciate the reviewer for your contribution to the peer review of this work. In the revised manuscript, we tried to explain the physical mechanisms more clearly.

1) We emphasized that the major role of IO (NA) warming on NA (IO) manifests through atmospheric teleconnection and added supporting figures.

2) We briefly mentioned the impact of oceanic processes on IO-NA warming in the "Discussion". The oceanic processes via ITF hardly contribute to IO warming by NA SST. Enhanced AMOC due to the IO-SST warming partly contributes to NA warming on multi-decadal timescales.

3) We added possible role of Pacific oscillation modes on IO-NA warming chain.

4) We checked citations for figures, references, and English grammar thoughtfully.

We believe that the revised manuscript provides better figures and more precise and clear physical mechanisms for IO-NA interaction by addressing your constructive comments. A detailed response to each comment is found below.

Major Comments

Question 1: Role of Pacific

The reviewer found that the role of the Pacific should be important in the Chain, as the authors also mentioned the atmospheric motion (ascending/descending) in the Pacific (Lines 94-127, Fig. S3). Analyses of this part are based on observational data. However, the authors do not give the model results to confirm this view. How are the model results in terms of the role Pacific?

Re: Thank you for the careful comments. We added the changes of simulated atmospheric motions associated with the IO/AO warming as the reviewer suggested (Fig. R1c-d), and we found that the climate model was able to reasonably simulate the observed features: “In the observation, the warm SST anomalies over the NA generated ascending motions and strong upper-level divergent flows, which are connected to upper-level convergences in the equatorial central and eastern Pacific regions (Fig. R1a). The descending motions connected to the upper-level convergence induced easterly anomalies over the central to western Pacific regions. Further, accumulated warm SST in the western Pacific likely strengthened the Walker circulation by enhancing the ascending motion over the equatorial western Pacific. The model reproduced observed patterns reasonably well but the divergence in the western Indian ocean is weaker than the observations.

Similarly, SST and surface wind anomalies associated with IO warming were observed to be linked to strong NA warming. The ascending motion over the Indian ocean generated sinking motion over Africa and propagate to the tropical Atlantic Ocean in the upper level and then induce ascending motion there with the NA warming. On the other hand, the rising motion over the IO generates a sinking motion over the Pacific region.”

Figure R1 Upper-level circulation induced by IO-NA warming. **a** Observed upper-level velocity potential (shading, $10^5 \text{ m}^2 \text{ s}^{-1}$) and divergent winds (m s^{-1}) anomalies regressed onto the NA index ($0^\circ\text{--}70^\circ\text{N}$, $80^\circ\text{W}\text{--}0^\circ$). The 11-year running average was used for 1950–2019, and the first and last 5 years were excluded. The dotted area represents significant velocity potential at the 95% confidence level. **b** Same as **a** but associated with the IO index ($35^\circ\text{S}\text{--}30^\circ\text{N}$, $40^\circ\text{--}120^\circ\text{E}$). **c** Same as **a** but from the model with observed NA index. **d** Same as **b** from the model with IO index. Long-term linear trends in the SST data were removed before regression.

On the other hand, how are the relationships between the Pacific indices (e.g., IPO, PDO, and decadal ENSO index) and IO-NA indices? Since tropical atmospheric motions are driven mainly by SST below, Fig. S3 may also suggest large changes of Pacific SST and the Pacific Walker Circulation. It is plausible that the three major oceans are in the same Warming Chain.

Re: To explore how the Pacific indices affect IO-NA interactions, we examined the lead-lag relationship between NA (or IO) and the Pacific oscillation modes (interdecadal Pacific oscillation, IPO; Pacific Decadal Oscillation, PDO; and decadal ENSO) using the observed data (1950–2020). The IPO is defined as the second empirical orthogonal function (EOF) (after the global warming mode) of decadal (11 years) low-pass filtered SST. PDO is defined as the leading principal component of North Pacific monthly sea surface temperature variability. The decadal ENSO is defined as the decadal (11 years) low-pass filtered Nino3.4 index. Noted that positive IPO pattern shows El Nino-like SST patterns. Within decadal time scale (~ 10 years), the peak of lead-lag correlation between the NA and PDO was found to occur at 3-5 lag years with a coefficient of 0.8–0.9 (Fig.R2a, blue line), indicating that NA leads to positive PDO

significantly with 3-5 lag years. Similarly, the NA leads the negative IPO phase with 6-10 years lag. The relationship between NA and decadal ENSO is relatively weaker than those of PDO and IPO. The IO leads both PDO and decadal ENSO at 3-5 lag years but there is no significant relation with IPO. These results showed that both IO and NA warming induces positive PDO with 3-5 lag years, suggesting that the IO-NA warming chain may interact with IO (NA) through PDO.

For the multi-decadal time scale, the positive IPO and decadal El Nino leads both IO and NA warming at 20 lag years, while IO and NA warming induces negative IPO and decal La Nina phase at 10-20 lag years. This lagged relationship may be attributed to the periodicity of the IPO and decadal ENSO. On the relationship with PDO, the NA leads negative PDO at 25 lag years but IO has no significant correlation with PDO at a multi-decadal time scale. These results showed that positive IPO may contribute to IO and NA warming with a lag of 20 years and on the contrary, the IO-NA warming could incudes negative IPO with a lag of 10-15 years, suggesting that IPO plays a role in the IO-NA warming chain with multi-decadal time scale. In summary, the Pacific indices could contribute to interdecadal modulation of the IO-NA warming chain, rather than a simultaneous feedback relationship.

Figure R2 Relationship between IO-NA warming chain and Pacific indices. a. Observed lead-lag correlation coefficient between the NA and PDO, IPO, and decadal ENSO indices. The lag is positive (negative) when the NA leads (lags). PDO is defined as the leading principal component of North Pacific monthly sea surface temperature variability. The IPO is defined as the second EOF (after the global warming mode) of decadally low-pass filtered SST (Yang et al. 2020). The decadal ENSO is defined as the Nino3.4 index of decadally (11year) low-pass filtered SST. The grey lines represent a 95% significance level and the green line shows zero. **b** Same as **a** but for IO index. The 11-year running

average data were used for 1950–2020 and long-term linear trends were removed. The last 5 years were excluded from the analysis and the long-term linear trends in the SST data were removed before regression.

Besides, the negative IPO phase pattern is unclear in Fig. 1a (Lines 98-99) compared to the typical IPO pattern (cf. Henley et al., 2015). Henley, B.J., Gergis, J., Karoly, D.J. et al. A Tripole Index for the Interdecadal Pacific Oscillation. Clim Dyn 45, 3077–3090 (2015).

Re: We agree with the reviewer's comments. We deleted those expressions in the revised manuscript.

Question 2: How the NA warming inducing the IO Warming

The authors do not clarify how the NA induces IO warming. In Line 165-167, they suggested that the IO warming caused by NA warming may also be attributed to oceanic processes associated with the ITF. BUT, the last of this paragraph, they wrote that “the ITF played a minor role in IO warming”. I am confused. What is the major factor in this part of the Warming Chain?

Re: We felt sorry for making the confusion. It has arisen from the unclear discussion about the role of ITF. We would suggest that the NA warming could induce the IO warming mainly by atmospheric teleconnection. The NA warming generates easterly anomalies over the Indian Ocean and western Pacific by modulated Walker circulation. The anomalous easterly over IO reduces mean-field westerly wind and latent heat flux, inducing IO warming.

The IO SST and surface wind changes induced by 1 °C NA warming largely resembled those observed in this study and showed significant IO warming (Fig. R3a). Warming also occurred in the northern and southern subtropical Pacific and equatorial western Pacific; however, relatively less SSTA warming prevailed in the eastern Pacific. The anticyclonic flows in the north and south Pacific is connected to the easterly anomalies in the western Pacific and the eastern Indian Ocean. As changes in the SST and surface winds by NA warming may affect the surface fluxes, we performed a surface heat budget analysis by computing the differences in the four surface heat flux components over the NA region for 1 °C warming (PI_NA+1C) and no warming (PI_NA+0C). The surface energy changes are mainly controlled by latent heat

flux (Fig. R3b), which is about two times greater than all other surface fluxes combined (Fig. R3c). Positive latent heat fluxes were consistent with the IO warming and thus contributed to IO warming (Fig. R3b). These results suggested that NA warming induces IO SST warming due to reduced wind-driven latent heat flux through enhanced Walker circulation in the Indo-Pacific.

The IO warming caused by NA warming may also be attributed to oceanic processes. The ocean heat advection from the western Pacific to the eastern IO through the Indonesian Throughflow (ITF) enhanced the warming effect over the IO. To verify this hypothesis, a mixed layer heat budget analysis for each advection term should be conducted. This issue will be studied in further studies. We briefly mentioned them in the “discussion” section.

Figure R3 Indian Ocean response by NA warming from pre-industrial simulations. (a) Horizontal patterns of SST anomalies ($^{\circ}C$), **(b)** latent heat fluxes ($W m^{-2}$) and surface winds ($m s^{-1}$), and **(c)** combined anomalies of sensible heat flux, solar and longwave radiation between PI_NA+0C and PI_NA+1C model experiments. 100-year simulated data are used for analysis.

Besides, in Fig. 2h, how should we understand “ocean upward water fluxes”? It is shown in cool color. Does that mean the negative effect of upwelling and ITF on IO warming? If so, how is it important for the ITF? Also, please explain where Fig. 2h is for citation.

Re: The ocean upward water fluxes mean upwelling in the sub-surface layer (0-50m). Negative anomalies mean relatively weakened upwelling. The warm water by ITF tends to be advected to the subsurface of the ocean but our simulation shows the subsurface water is cooled in the experiment with PI_NA+1C relative to the PI_NA+0C, suggesting that ITF did not contribute to IO warming. We removed this figure and related explanation because they are not important

to show the main idea of this study and not to make unnecessary confusion.

Moreover, since the correlation analyses (Fig. 1e) suggest the simultaneous coefficient is the largest, does this mean the atmospheric forcing is the major cause? Because the oceanic process needs time to respond (as seen in the AMOC strengthened in Fig. 2e). If so, how does the atmospheric forcing related to the NA warming affect the IO warming?

Re: Thank you for the constructive comments. We argued that the NA warming could warm up SST over the Indian ocean by atmospheric teleconnection with reduced latent heat flux during all periods.

To analyze the change of atmospheric circulation by NA warming, we examined the change of Walker circulation at the equator between PI_NA+1C and PI_NA+0C. The NA warming pattern resembled a developing positive phase of the Atlantic Multidecadal Oscillation (AMO), and subsequent strong warming was observed in the tropical IO. The warm SST anomalies over the NA generated ascending motions and strong upper-level divergent flows, which are connected to upper-level convergences in the equatorial central and eastern Pacific regions (Fig. R4a). The descending motions dominate in this region. Additionally, descending motions in the central Pacific induced easterly anomalies over the central and western Pacific regions. Further, warmer SST in the western Pacific strengthened the Walker circulation by enhancing the ascending motion over the equatorial western Pacific. The Walker circulation generates easterly anomalies in the Indian Ocean and western Pacific, which can be simultaneously induced by the eastward propagation of Kelvin waves from the Atlantic as well. As a result, reduced mean wind speed and associated latent heat flux enhance IO warming (Fig. R3).

Figure R4 Change atmospheric circulation. **a** Walker circulation changes (arrows) and troposphere vertical velocity anomalies (color shading) averaged between 5° S and 5° N over between PI_NA+1C and PI_NA+0C. The vertical velocity is magnified by a factor of 750 to make its scale comparable to that of zonal wind. **b** Same as **a** but between PI_IO+1C and PI_IO+0C.

We suggested that the IO warming could warm up SST over NA by atmospheric teleconnection with enhanced longwave radiation during all periods. We also found that the oceanic processes (i.g. AMOC) partly contribute to NA warming after 40-50 years from the initial time. In summary, the major cause of NA warming is atmospheric surface forcings and ocean processes partly contribute to NA warming during the final phase.

To understand how IO warming affected the NA SST by atmospheric teleconnection, we examined differences in the temporal evolution of surface heat fluxes between PI_IO+0C and PI_IO+1C. Note that surface heat fluxes correspond to the impact of IO on NA through atmospheric teleconnection during most simulation periods. Figure R5a depicts that NA SST increased rapidly, upon IO warming, for approximately the initial 20 years; a constant increase in the NA SST was observed for 50 years. The surface net heat flux was positive for all the periods (Fig. R5c). The net surface flux adjusted very quickly, within a span of a few years, but slowly decreased with time. The longwave radiative flux over NA increased during the initial 20 years and then decreased slowly (Fig. R5b). This suggested that the net surface flux played a major role in the temporal evolution of SST during the initial period.

We also examined how the ocean processes affect NA SST by IO warming. AMOC showed no significant changes until 40 years but it gradually increased for 100 years (Fig. R5d), suggesting that the AMOC partly contribute to NA warming during the later phase (after 40-50 years). The changes in AMOC were attributed to atmospheric teleconnection generated by IO warming. The sea surface salinity (SSS) increased over the tropical Atlantic Ocean and the enhanced SSS strengthened indicating that IO warming enhanced the AMOC with multi-decadal timescales, contributing to NA SST warming after 50–100 years. These results imply that the NA response due to IO warming resulted from mainly atmospheric teleconnection and ocean circulations partly affect only during later phase. To avoid confusion, we deleted the role of the oceanic processes in the main text because they are not the main cause and we briefly mentioned them in the “discussion section”.

Figure R5. IO-NA interaction from pre-industrial simulations. a-e NA response by IO warming. a Temporal evolution in annual-mean SST anomalies ($^{\circ}\text{C}$) over NA region. **b** Same as **a** but for longwave radiation. The downward direction is positive (warming). **c** Same as **a** but for net surface fluxes. The blue (or red) line shows annual mean data and the black line represents an 11-year running mean of the red line (or blue line). **d** Temporal evolution of anomalies in Atlantic meridional overturning circulation (AMOC) strength. The AMOC strength is estimated as the maximum stream function within 500–5500 m, 30°N to 70°N . The red (or blue) line shows annual mean data, and the black line shows an 11- year running mean of the red (or blue) line.

Question 3: Mechanism verified by model results

In the model results, the authors showed sufficient evidence that there is a Warming Chain between the NA and the IO, in which the changes of SST in both oceans are in a similar linear change (Figs. 3-4). However, physical processes can be further explored using the model results since the observational data are short.

One concern is what leads to the surface flux change in the IO-NA (Fig. 2). For example, the Rossby wave chain or the global Walker Circulation change may contribute to the IO-NA warming. So far, the authors just showed the final results rather than the processes behind them. At least, they can verify Fig. S3 by the model results.

Re: To analyze the change of atmospheric circulation by NA warming, we examined the change of Walker circulation at the equator between PI_NA+1C and PI_NA+0C. The NA warming pattern resembled a developing positive phase of the Atlantic Multidecadal Oscillation (AMO), and subsequent strong warming was observed in the tropical IO. The warm SST anomalies over the NA generated ascending motions and strong upper-level divergent flows, which are connected to upper-level convergences in the equatorial central and eastern Pacific regions (Fig. R4a). The descending motions dominate in this region. Additionally, descending motions in the central Pacific induced easterly anomalies over the central and western Pacific regions. Further, warmer SST in the western Pacific strengthened the Walker circulation by enhancing the ascending motion over the equatorial western Pacific. The Walker circulation generates easterly anomalies in the Indian Ocean and western Pacific, which can be simultaneously induced by the eastward propagation of Kelvin waves from the Atlantic as well. As a result, reduced mean wind speed and associated latent heat flux enhance IO warming (Fig. R3).

The enhanced rising motion by the IO surface warming generated sinking motion over the central and eastern Pacific and intensifies the Indo-Pacific Walker circulation (Fig. R4b). The enhanced convection induces partly sinking motion over Africa and propagates to the troposphere of the tropical Atlantic Ocean, with sinking motion in the lower level and ascending motion in the middle and higher level. Meanwhile, the increased cloudiness over western tropical Atlantic to NAO by IO warming (Fig. R6a) contribute to surface warming by enhanced longwave radiation to the ocean (Fig. R6b). These results showed that the IO-NA warming chain is dominated by atmospheric teleconnection.

Figure R6 North Atlantic Ocean response by IO warming from pre-industrial simulations. (a) Horizontal patterns of cloudiness (%) and **(b)** longwave heat fluxes (W m^{-2}) between PI_IO+0C and PI_IO + 1C model experiments. 100-year simulated data are used for analysis.

Other minor comments

It seems that there is no content for mixed-layer heat budget analysis in the revision, but it is listed in the method.

Re: We removed it in the revised manuscript.

Citations for Figure 4 are all wrongly replaced by Figure 2. (See section IO-NA warming chain evidenced through historical simulations)

Re: We checked all citations for figures and corrected them in the revised manuscript—“The IO Index ensemble simulated by the observed SST anomalies over the NA region (HIS_OBS_NA) captured the observed trends reasonably well for most of the periods (Fig. 4a). The range of mean for each member from the ensemble was ± 0.12 °C (approximate biases of 10%–15%), indicating that the impacts of internal variability on the simulated IO SST were moderate; thus, the response of the IO was influenced by NA warming was realistic. We re-conducted the same historical simulation, except for the prescribed climatological SST of NA (HIS_CLIM_NA), to eliminate the impact of NA warming on IO warming. The results showed that IO SST warming was reduced by 40–50% (green line in Fig. 4a), indicating that the observed IO warming arose from the remote effects of NA as well as local IO warmings.

For another historical simulation with observed SST forcings over the IO region (Fig. 4b), the mean SST ensemble over the NA region depicted a realistic temporal evolution, with negative SST anomalies being observed from the 1970s to the 1990s along with rapidly increasing positive SST anomalies after the late 1990s. It was slightly less than that observed, which may be attributed to the systematic mean SST biases of the south of Greenland³⁴. The ensemble spread from each member was moderate ($\pm 0.26^{\circ}\text{C}$) signifying that all ensemble simulations captured the NA warming trends induced by IO warming reasonably well. The same historical simulation, but with climatological SST nudged over the IO region, yielded a 50% reduction in NA SST warming (green line, Fig. 4b). This result suggested that approximately half of the NA warming may be attributed to IO warming.

We further examined the sensitivity of the IO (or NA) response to NA (or IO) warming. For this, we conducted two additional groups of model experiments. The first involved historical simulations with observed IO SST patterns and trends ($0.01\text{--}0.70^{\circ}\text{C}$ per 70 years). Different trends were obtained by multiplying different scale factors with the observed values (Supplementary Fig. 8a). The second involved specifying different trends of observed SST anomalies over the NA region ($0.01\text{--}1.02^{\circ}\text{C}$ per 70 years) (Supplementary Fig. 8b). The model was freely coupled in other areas, and external forcings were based on the CMIP6 historical simulation protocol.

Historical runs that nudged various IO (or NA) SST warming (Figs. 4c and 4d) demonstrated that increasing IO (or NA) SST could lead to a warmer NA (or IO). The IO SST trend forced by the NA SST was smaller than the NA SST trends (by approximately 40–50%). Thus, IO warming may be attributed to a remote NA impact and local surface flux changes. We will study this issue in more depth in our future studies.”

References should be listed correctly. (See No. 3 and 21, no published year; No. 7 and 16, duplication; other many spelling errors.)

Re: We checked all reference lists in the revised manuscript.

3. Hansen, J., Ruedy, R., Sato, M., & Lo, K. Global surface temperature change. *Rev. of Geophys.*, **48**, RG4004 (2010).

7. Delworth, T. L. et al. The North Atlantic oscillation as a driver of rapid climate change

in the Northern Hemisphere. *Nat. Geosci.* **9**, 509–512 (2016).

16. McGregor, S. et al. Recent Walker circulation strengthening and Pacific cooling amplified by Atlantic warming. *Nat. Clim. Change* **4**, 888–892 (2014).

21. Xie, T., Li, J., Chen, K. et al. Origin of Indian Ocean multidecadal climate variability: role of the North Atlantic Oscillation. *Clim. Dyn.* **56**, 3277–3294 (2021)

References

Yang, Y.-M., An, S.-I., Wang, B. & Park, J. H. A global-scale multidecadal variability driven by Atlantic multidecadal oscillation. *Natl Sci. Rev.* **7**, 1190–1197 (2020).

Reviewer #2 (Remarks to the Author):

The authors have addressed my concerns in this revision. They have explained the interaction between the North Atlantic (NA) and the Indian Ocean (IO), in which the atmospheric processes related to the NA play a major role in the IO warming. This explanation sounds reasonable since the warming effect induced by surface heat flux would be faster than the ocean dynamics. They also discussed the Pacific effect on the NA-IO warming chain, while the reviewer still thinks this could be a systematically global interaction that the NA, the IO, and the Pacific are all connected as the authors responded to Question 1 Role of Pacific (also see Line 245-271 in the main text). In other words, that would be a warming chain for NA-IO-Pacific. Plausible evidence for such a global chain is given by the change of the Walker Circulation, as shown in Figure S3 (or Figure R1 for my review). In short, I recommend this study for publication, and the authors could further consider the issue of the global chain associated with NA-IO-Pacific.

RESPONSE TO REVIEWER COMMENTS

Reviewer #2 (Remarks to the Author):

The authors have addressed my concerns in this revision. They have explained the interaction between the North Atlantic (NA) and the Indian Ocean (IO), in which the atmospheric processes related to the NA play a major role in the IO warming. This explanation sounds reasonable since the warming effect induced by surface heat flux would be faster than the ocean dynamics.

They also discussed the Pacific effect on the NA-IO warming chain, while the reviewer still thinks this could be a systematically global interaction that the NA, the IO, and the Pacific are all connected as the authors responded to Question 1 Role of Pacific (also see Line 245-271 in the main text). In other words, that would be a warming chain for NA-IO-Pacific. Plausible evidence for such a global chain is given by the change of the Walker Circulation, as shown in Figure S3 (or Figure R1 for my review).

In short, I recommend this study for publication, and the authors could further consider the issue of the global chain associated with NA-IO-Pacific.

Re: We thank the reviewer for your constructive and valuable comments on the peer review of this work. We appreciate it.

In the revised manuscript, we added the role of the Pacific effect on the NA-IO warming chain in the “Abstract” section-“the Pacific indices could contribute to interdecadal modulation of the IO-NA warming chain”

Also, the possibility of systematic global interaction that IO, NA, and Pacific all are connected through atmospheric teleconnection was added in the “Discussion” section- “We discussed the Pacific effect on the NA-IO warming chain, which could shed light on systematical global interaction that the NA, the IO, and the Pacific are all connected through the change of the Walker Circulation (Supplementary Figure S3). We will consider the issue of the global chain associated with NA-IO-Pacific in further study”.